# Updated-Food Choice Questionnaire: Cultural Adaptation and Validation in a Spanish-Speaking Population from Mexico

**DOI:** 10.3390/nu16213749

**Published:** 2024-10-31

**Authors:** Miguel Amaury Salas-García, María Fernanda Bernal-Orozco, Andrés Díaz-López, Alejandra Betancourt-Núñez, Pablo Alejandro Nava-Amante, Ina Danquah, J. Alfredo Martínez, Daniel A. de Luis, Barbara Vizmanos

**Affiliations:** 1Doctorado en Ciencias de la Nutrición Traslacional, Departamento de Clínicas de la Reproducción Humana, Crecimiento y Desarrollo Infantil, División de Disciplinas Clínicas, Centro Universitario de Ciencias de la Salud (CUCS), Universidad de Guadalajara (UdeG), Guadalajara 44340, Mexico; amaury.salas@alumnos.udg.mx (M.A.S.-G.); alejandra.bnunez@academicos.udg.mx (A.B.-N.); pablo.nava@alumnos.udg.mx (P.A.N.-A.); 2Laboratorio de Evaluación del Estado Nutricio, Departamento de Clínicas de la Reproducción Humana, Crecimiento y Desarrollo Infantil, División de Disciplinas Clínicas, Centro Universitario de Ciencias de la Salud (CUCS), Universidad de Guadalajara (UdeG), Guadalajara 44340, Mexico; 3Instituto de Nutrigenética y Nutrigenómica Traslacional, Departamento de Biología Molecular y Genómica, División de Disciplinas Básicas, Centro Universitario de Ciencias de la Salud (CUCS), Universidad de Guadalajara (UdeG), Guadalajara 44340, Mexico; 4Doctorado en Ciencias de la Salud Pública, Departamento de Salud Pública, División de Disciplinas para el Desarrollo, Promoción y Preservación de la Salud, Centro Universitario de Ciencias de la Salud (CUCS), Universidad de Guadalajara (UdeG), Guadalajara 44340, Mexico; 5Nutrition and Mental Health Research Group (NUTRISAM), Universitat Rovira i Virgili (URV), 43204 Reus, Spain; andres.diaz@urv.cat; 6Institut d’Investigació Sanitària Pere Virgili (IISPV), 43204 Reus, Spain; 7CIBERobn Physiopathology of Obesity and Nutrition, Institute of Health Carlos III (ISCIII), 28029 Madrid, Spain; jalfredo.martinez@imdea.org; 8Centro de Investigación Educativa y Bienestar Universitario, Departamento de Disciplinas Filosófico, Metodológico e Instrumentales, División de Disciplinas Básicas, Centro Universitario de Ciencias de la Salud (CUCS), Universidad de Guadalajara (UdeG), Guadalajara 44340, Mexico; 9Heidelberg Institute of Global Health (HIGH), Heidelberg University Hospital and Medical Faculty, Heidelberg University, 69120 Heidelberg, Germany; ina.danquah@uni-heidelberg.de; 10Centro de Investigación de Endocrinología y Nutrición Clínica, Universidad de Valladolid, 47005 Valladolid, Spain; dluisro@saludcastillayleon.es; 11Precision Nutrition and Cardiometabolic Health Program, Research Institute on Food and Health Sciences IMDEA Food, UAM + CSIC, 28049 Madrid, Spain

**Keywords:** food choice questionnaire, food selection, validity, reliability, factor analysis

## Abstract

Background: Determinants and motives related to food selection have evolved in a globalized and changing world. The traditional and useful Food Choice Questionnaire (FCQ), created in 1995, needs to be updated, adapted to new scenarios, and validated. Objectives: This study aimed to: (1) assess face validity (FV) of the original 36-item FCQ, (2) generate an Updated-FCQ (U-FCQ) and assess its content validity (CV) (instrument suitability), and (3) evaluate its construct validity and reliability in a Spanish-speaking population from Mexico. Methods: FV involved a panel of nutrition professionals (NPs) rating the original items’ clarity, relevance, specificity, and representativeness. A literature review process updated the FCQ by adding new items. CV with a second NP panel allowed calculating content validity ratio (CVR). Construct validation was performed via exploratory and confirmatory factor analysis (EFA-CFA). Internal consistency through Cronbach’s alpha (CA) and test–retest reliability via intra-class correlation (ICC) were assessed. Results: The FV (n = 8) resulted in the modification of 11 original items. The literature review added 36 new items (15 from previous adaptations and 21 original items). The CV (n = 13) identified nine items (non-acceptable CVR), prompting reformulation of seven and removal of two. The NPs’ feedback added six new items. The EFA-CFA (n = 788) developed a 75-item U-FCQ with eight dimensions: sensory appeal, mood, health and nutritional content, price, food identity, environmental and wildlife awareness, convenience, and image management. CA ranged from 0.74–0.97 (good–excellent) and ICC from 0.51–0.78 (moderate–good). Conclusions: This study provides a useful instrument for the assessment of food choices and lays the groundwork for future cross-cultural comparisons, expanding its applicability in wider settings.

## 1. Introduction

Human food choices are complex and determined by the interactions of various socioecological factors [1]. These factors act on five levels: (1) individual characteristics (psychological and biological aspects, knowledge of health and nutrition, food preferences and sensory characteristics); (2) interpersonal characteristics (family, culture, social context, etc.); (3) environment (area of residence, food marketing, price and availability of food, time for food preparation, etc.); (4) food-related public policies (dietary guidelines and food legislation); and (5) agri-food sustainability (concern for the integrity and maintenance of the environment, as well as the appropriate use of natural resources) [1,2,3,4,5].

These complex determinants collectively shape an individual’s food choices, thereby playing a pivotal role in framing the quality of their diet. In turn, diet quality can significantly influence their susceptibility to developing comorbid diseases, such as overweight/obesity, type 2 diabetes, hypertension, and cancer, among others [6]. Consequently, understanding this phenomenon holds substantial implications, ranging from private nutritional consultation, development of innovative food products, and, of course, the planning of public health policies and strategies aimed to promote and supervise the development of healthier eating habits [7].

Among the most notable tools for analyzing the determinants of food choices is the Food Choice Questionnaire (FCQ), developed in 1995 by Steptoe and collaborators [8]. This instrument evaluates individual food choice motives and dimensions. In its original version, 36 items are distributed in nine dimensions: health, natural content, weight control, familiarity, ethical concerns, sensory appeal, price, convenience, and mood. This questionnaire has been adapted and validated for different populations and contexts [9,10,11,12]. In fact, to date, more than 30 modified versions of the FCQ have been developed, mostly in English-speaking countries. In Mexico’s case, the FCQ was previously used, with some adaptations, aiming to identify food choice motives in Mexican consumers [13] and, specifically, in organic foods consumers [14]. Nevertheless, several emerging reasons for food choices have been overlooked, such as the restriction of certain nutrients (sodium, gluten, artificial sweeteners, etc.) [15], the importance attached to food policies and nutritional labeling [16], the perceived value of foods or impressions created by food choices [17], among others. As a result, it is necessary to incorporate new items into this questionnaire, items that evaluate emerging aspects, and to adapt it to the cultural context where it will be applied.

In addition, although the FCQ has been widely studied, very few authors have implemented a structured approach to adapt this tool to the cultural particularities of the context or population where it has been applied [7]. Moreover, most of the existing FCQ adaptations only report the construct validity, defined as the degree to which a test, including its items, adequately represents or fits the content of its domains [18,19]. However, they do not report the face validity (the evaluation of the items’ appropriateness perceived by the potential users of the tool) [20] and the content validity (to assess if the items adequately measure the domain of interest) [19,21].

Considering that both face and content validity have usually been neglected in research [22], the emphasis on the methodology and the approach to these processes could contribute to the development of more appropriate tools that offer an accurate reflection of what is intended to be measured. Such considerations are especially true when the aim is to validate a tool not only in another language but also in a different temporal and cultural setting [21]. Adaptation is essential, since idiosyncrasies and food intake are different between countries, including those within Spanish-speaking populations.

In response to this evolving landscape of food choice motives, we undertook the task of developing an Updated-Food Choice Questionnaire (U-FCQ) and to validate it. Therefore, the objectives of this study were: (1) to evaluate the face validity of the FCQ, (2) to update the FCQ (U-FCQ) and to assess its content validity and suitability, and (3) to evaluate the construct validity and reliability of the U-FCQ in a Spanish-speaking population from Mexico.

## 2. Materials and Methods

### 2.1. Study Design

A sequential mixed-method research approach and a cross-sectional study design were used to develop and test the validity and reliability of the FCQ, comprising several steps. For cross-cultural adaptation and validation approach, we followed recommendations from Boateng and colleagues [19] and Gjersing and colleagues [23]. These include the translation methodology, the use of committees involving experts or professionals in the field, literature review, and the exploratory and confirmatory analyses on Mexican participants. In accordance with this, the validation study included the following stages: (1) translation and face validation, (2) literature review, (3) content validation, and (4) construct validation.

### 2.2. Translation and Face Validation

In order to evaluate how the original 36-item FCQ [8] performed in the Mexican population and whether it was appropriate for its current context, this tool was translated to Spanish (for the Mexican context). The translation and back-translation method was applied by the first author of this paper (M.A.S.-G.), whose native languages include both Spanish and English. The translation was reviewed independently by two team members fluent in English (M.F.B.-O. and B.V.). Each FCQ item is introduced with the following statement: “It is important to me that the food I eat on a typical day…”. Then, the participant is asked to evaluate the importance given to each sentence according to a 4-point Likert-type scale, going from 1 (not at all important) to 4 (very important).

Then, a face validity assessment was carried out. According to the methods published by Trakman and colleagues for the development and validation of nutrition questionnaires [24], Spanish-speaking nutrition professionals (NPs) from the Health Sciences Center (*Centro Universitario de Ciencias de la Salud*—CUCS for its Spanish acronym) from the University of Guadalajara (*Universidad de Guadalajara*), in Mexico, were consulted using a structured approach. Face validity assessed the degree of appropriateness of each item (if it was clear, relevant, specific, and representative), according to the perception of the NPs. Additionally, the NPs could include comments or suggestions that could be considered by the research team. A detailed description of the face validation methods is presented in Appendix A.

### 2.3. Literature Review

In parallel with conducting the face validation phase, we conducted a literature review. After deeply reviewing the relevant factors for food choices (individual characteristics, interpersonal characteristics, environmental characteristics, food-related public policies, and agri-food sustainability), it was identified (by the NPs and the authors) that the original FCQ did not contemplate all the elements currently described in the literature for food choices. Consequently, it was decided to conduct a more specific literature review from August to December 2021 to identify new items and dimensions of previously adapted and validated versions of the FCQ. We aimed to develop an updated FCQ (U-FCQ) that would assess a wider range of currently relevant aspects related to food choices. Therefore, a PubMed and Web of Science “all fields” search was performed (in English), using the following keywords in combination with the Boolean operators: [“Food Choice Questionnaire” OR “FCQ”] AND [“Adaptation” OR “Content validation” OR “Validity”]. All English, Spanish, and Portuguese language adaptations were included for this purpose. Furthermore, new items/dimensions were also proposed directly by the research team according to the previous review of factors determining food choices.

### 2.4. Content Validation

Once the FCQ was updated through the literature review and continuing with the use of experts or professionals in the field methodology, a second panel of NPs was formed. Participants were considered from a group of academics who are regularly consulted for various topics by the Coordination of the Nutrition Bachelor’s degree program from the CUCS, University of Guadalajara. Additionally, members of the first panel were also invited to participate. The objective of this panel was to evaluate whether each one of the items (original and new ones) was clear, representative, relevant, and specific. If the NPs suggested adding a new item, the authors deliberated whether or not to include it, based on the literature review and their own research experiences. For further information, see Appendix A.

### 2.5. Construct Validation

The construct validation of the U-FCQ was conducted by a cross-sectional validation study design among Mexican adults 18 years of age and older from two different research projects. Both projects were conducted according to the guidelines laid down in the Declaration of Helsinki and were approved by the Research, Research Ethics and Biosafety Committees from CUCS, University of Guadalajara (*Universidad de Guadalajara*), in Mexico. The first project aimed to analyze the risk factors for food insecurity and its consequences on lifestyle and health among university students and workers in the metropolitan area of Guadalajara, Mexico (approval advice code CI-02323). The second project was specifically developed for the construct validation of this tool, which included participants from different Mexican states (approval advice code: CI-04023).

Regarding sample size calculation, although there is no consensus for determining the number of participants needed to perform factor analysis, it has been suggested to include at least 10 participants per item present in the tool to be validated [25]. Accordingly, considering the existence of 76 items after the abovementioned processes of face and content validation, the minimum sample size was calculated to be 760 participants.

Participants in the first project were recruited face-to-face (direct visits to university students’ classrooms and to university workers in their usual work areas), while for the second project, they were recruited virtually through a flyer published on social media. Informed consent was obtained from all participants involved in the construct validation study. For both projects, volunteers were asked to answer online self-administered questionnaires to obtain the following data: sociodemographics (country and area of residence, educational level, socioeconomic and marital status), self-reported weight and height, food consumption quality, food safety status, and the U-FCQ.

The inclusion criteria for the university students and workers project were being a student or university worker and 18 years of age or older. On the other hand, due to the fact that, in that project, the presence of metabolic syndrome and its components were determined, subjects with characteristics that could cause alterations in these were excluded, such as: pregnant or breastfeeding women, consumption of corticosteroids, isotretinoin, antiretrovirals, or androgens, and individuals diagnosed with cancer or those who received treatment for this condition in the last 6 months. For the second project, Mexican men and women 18 years of age or older from the general population who had access to a computer or electronic device were invited to complete the surveys. Participants who had previously answered the U-FCQ were excluded.

To determine the socioeconomic status (SES), we used the *Asociación Mexicana de Agencias de Inteligencia de Mercado y Opinión* (AMAI) tool, validated for the Mexican population [26]. Seven socioeconomic levels are defined, using letters: A/B, C+, C, C−, D+, D, and E (from highest to lowest). These levels were then grouped into two categories for statistical analysis purposes: high socioeconomic status (A/B and C+) and low–middle socioeconomic status (all the other categories).

Body mass index (BMI) was determined using height and weight measured according to the ISAK protocol [27] for participants from the first project and self-reported height (in cm, and then converted to meters) and weight (in kilograms) for participants from the second project. For calculations, we used the formula weight divided by height squared (kg/m^2^). Then, for statistical analysis purposes, BMI was classified into normal weight (<25 kg/m^2^) or overweight/obese (≥25.0 kg/m^2^) [28].

To assess the food security status of the participants, they were asked to answer the validated Latin American and Caribbean Scale of Food Security (its acronym in Spanish: ELCSA) [29]. According to affirmations related to lack of money or other resources a score is obtained, and the participants are classified as having food security, mild food insecurity, moderate food insecurity, or severe food insecurity.

The quality of food consumption was assessed using the validated second version of the Mini-Survey to Evaluate Food Intake Quality (its acronym in Spanish: Mini-ECCA v.2) [30]. It includes 14 items that evaluate the consumption of natural water, vegetables, legumes, sweetened beverages, and processed foods and the types of cereals and meats consumed. Additionally, it includes photographs that facilitate the estimation of the amount of food ingested [30]. Based on equations, it allows categorization of participants into three food intake patterns: “Healthy Food Intake”, “Habits in Need of Improvement”, or “Unhealthy Food Intake”.

Finally, the U-FCQ obtained after the face and content validation was applied to assess the food choice motives of participants. Respondents were instructed to rate the importance of each item according to their food choices on a 4-point scale (1 = not at all important to 4 = very important). Fifteen days after the first survey, participants received, via email, their results and personalized feedback regarding their BMI, food security status, and food consumption quality. Additionally, the participants were sent a new survey to complete the U-FCQ a second time. After completing this second survey, they also received personalized information related to their reasons for choosing food.

### 2.6. Statistical Analysis

Quantitative variables are presented as mean (standard deviation), while categorical variables are presented as frequency (percentage). Comparison of the mean of age and BMI between men and women was analyzed using the *t*-test. Comparison of qualitative variables between sexes was performed using the chi-square test.

For face validity, the frequency and percentage of compliance with the criteria (clear, representative, specific, and relevant) were calculated for each item. For the content validity, we calculated the content validity ratio (CVR) for each item, according to Lawshe et al.’s formula [31]:CVR = [ne − (N/2)]/(N/2),(1)
where “N” is the total number of the participants who evaluated the item, and “ne” is the number of participants in agreement with the item. For calculations, an Excel template available online was used [32]. The value of CVR ranges from 0 to 1, where 1 represents the maximum degree of agreement among the NPs. Lawshe et al. [31] previously determined the minimum CVR value required to maintain the items in the tool, according to the number of assessors included. This was considered to decide the inclusion/exclusion of specific items, after discussion with the research team.

For the construct validation of the U-FCQ, an exploratory factor analysis (EFA) on the 76 items was carried out through principal component analysis (PCA), with varimax rotation to ensure that the principal components remained uncorrelated and, thus, facilitate the interpretation. A priori, the Kaiser–Meyer–Olkin (KMO) test was applied to evaluate the adequacy of the model (acceptable value must be equal to or greater than 0.70) [25]. Additionally, Bartlett’s test of sphericity (*p* value < 0.05 considered significant) was used to determine the sufficiency of correlations among variables to carry out the factor analysis [33]. For extraction, the resulting factors/dimensions were required to meet the following criteria: eigenvalues ≥1, contain at least 3 items with factor loadings ≥0.30, and have internal logic within items of a dimension [25]. Finally, factors/dimensions were labeled according to the characteristics of their items.

The resulting structure was tested through confirmatory factor analysis (CFA). Chi-square values, degrees of freedom, root mean square error of approximation (RMSEA) (value < 0.08 indicative of a good fit), comparative fit index (CFI) (≥0.95 indicative of good fit), Tucker–Lewis index (TLI) (≥0.95 indicative of good fit), and standardized root mean squared residual (SRMR) (value < 0.08 indicates good fit) were used for evaluating the model fit [34,35,36], for the obtained model and the two adjacent factors models. Also, correlation coefficients among the factors were analyzed. Discriminant validity was analyzed using the Fornell and Larcker criterion [37], where average variance extracted (AVE) for every factor must be greater than the shared variance (SV) of the correlation coefficient between factors [38,39]. Moreover, for the same purpose, we applied the proposed approach of assessing the correlation between factors [39]. Coefficients below 0.85 determine discriminant validity [40].

For reliability, internal consistency was assessed using Cronbach’s alpha coefficients for the whole U-FCQ and for each one of its dimensions (stratified Cronbach’s alpha); coefficients were classified as excellent (≥0.90), good (0.80–0.89), acceptable (0.70–0.79), questionable (0.60–0.69), poor (0.50–0.59), and unacceptable (<0.59) [41]. Also, the intra-class correlation coefficient (ICC) was employed to assess reliability in the importance given to each item and dimension between test and retest (from 1, not important at all, to 4, very important). The ICC’s importance scores were interpreted as poor (<0.50), moderate (0.50–0.75), good (0.75–0.90), and excellent (>0.90) [42]. In addition to the ICC, test–retest reliability was also assessed through Pearson’s correlation coefficients, which were then interpreted as weak (≤0.39); moderate (0.40–0.69); strong (0.70–0.89); and very strong (0.90–1.00) [43].

All statistical analyses were performed using STATA^®^ software, version 15.0 (StataCorp, College Station, TX, USA). A *p*-value < 0.05 was considered as statistically significant.

## 3. Results

### 3.1. Translation and Face Validation

A first panel of eight NPs evaluated the face validity of the original version of the FCQ (36 items distributed into nine dimensions). In this panel, there were two men and six women, with a median age of 30 years, all with a bachelor’s degree in nutrition sciences and three with a science master’s degree. For more information about the NPs’ characteristics, see Appendix A.

Most of the observations from this panel were focused on adapting words to the Spanish-speaking Mexican context. Thus, 11 items were modified in their wording to facilitate their understanding. For example, the item “Is easy to prepare” was changed to “Is practical (easy preparation and almost immediate consumption)”, in order to cover the reasons for selecting foods that do not necessarily require cooking skills, like blending, toasting, mixing ingredients, using the microwave, among others, and that do not automatically imply the consumption of processed foods. This was performed because most of the NPs (six out of eight) considered this item to be repetitive, since it was very similar to another original item “Can be cooked very simply”.

On the other hand, the item “Can be cooked very simply”, was changed to “Can be easily cooked”, to consider the ease of the cooking process. Finally, the item “Takes no time to prepare”’ was considered repetitive with respect to “Is easy to prepare” by most of the participants (five out of eight), therefore, it was changed to “Can be cooked quickly”. The rest of the items not presented remained the same as in the original version (25 out of 36). Full descriptions of the modifications made in this stage are presented in Appendix A.

### 3.2. Literature Review

As part of the methodology for updating the questionnaire, 33 previous FCQ adaptations were identified and analyzed by the research team in order to detect emerging food choice motives that have been assessed since its creation. Among these, 15 articles were selected [12,44,45,46,47,48,49,50,51,52,53,54,55,56,57], from which 15 items were considered important and relevant to the current Spanish-speaking Mexican culture context and were included in the questionnaire. Additionally, noteworthy aspects related to food selection were identified, which have not been evaluated through the FCQ versions so far. As a result of this particular search and from the research team deliberation, 21 items were originally developed and proposed by the research team to evaluate the following novel food choice motives: ease of digestion, energetic contribution of food and nutritional requirements, freshness, carbohydrate content, homemade preparation, attractiveness of its advertisement/package, the capacity of the food to induce satiety, avoidance of gluten, similarity to the traditional Mexican diet, adherence to Mexican dietary guidelines, among others.

Out of the 36 new items (15 derived from previous adaptations and 21 developed by the research team), 20 items were added to the nine original dimensions, whereas 16 items were distributed among three new proposed theoretical dimensions: “image management” (5 items), “sustainability” (8 items) and “food policies/legislation” (3 items). The addition of these new dimensions and items resulted in an updated 72-item FCQ. For a complete description of the tool at this phase, and each item’s source, see Table 1. It is important to notice that items are presented in Spanish, as they were validated, and in English, as a suggested translation.

### 3.3. Content Validation

For the content validation phase of the 72-item FCQ, we included 13 NPs: 7 of them were from the group of consultants mentioned in the Section 2, 5 were from the first panel (5 out of 8), and 1 was an external collaborator of the research team. The median age was 35 years; eight of them had postgraduate studies (master’s or PhD), and four of them were full-time professors, with a median time working at the university of two years. For further detail on the NPs’ characteristics of this second panel, see Appendix A.

According to the number of experts involved (n = 13), a minimum CVR of 0.54 was required to maintain an item in the questionnaire [18]. The CVR of the tool items ranged from 0.07 to 1. Absolute agreement among NPs was found for 22 items (30.6%). Furthermore, there was highly significant agreement for 41 items (56.9%) with a CVR from 0.54 to 0.84, while 9 items (12.5%) did not meet the minimum CVR required. To see the details of the number of NPs in disagreement and the CVR calculated for each item, see Table 1.

Subsequently, the items that failed to obtain the minimum required CVR value (items with superscript 5 in Table 1) were reevaluated along with the NPs to identify the reason why they were considered not clear, relevant, specific, or representative. Upon further analysis, we found that the main reason was lack of clarity. In this sense, the NPs suggested changing their wording (seven out of nine items), as well as including examples of the terms used in the items, e.g., allergies, stomachache, diarrhea, etc. for physical discomfort.

As for the item “Is produced without human exploitation”, NPs in disagreement (sic out of six) considered that the evaluation of this concept is complicated, causing this item to be unclear. Therefore, it was decided to remove it from the tool. Only the item “Comes from countries I approve of politically” was considered by four out of six NPs in disagreement. Thus, this second item was also eliminated (2 of 9 items with low CVR value; obtaining a 70-item questionnaire at that moment). For more information on the assessment of these items, see Appendix A.

Furthermore, NPs suggested broadening the assessment of food choice motives through the incorporation of the following six items: “Is low in salt”, “Is free of non-caloric sweeteners, for example: sucralose, stevia, etc.”; “Is on sale”, “Is produced in Mexico”, “Has the shortest storage time (as fresh as possible)”, and “Is consumed by most of the members of my household”. Following an evaluation by the research team, it was decided to add them to the tool. As a result, at this stage a 76-item questionnaire was obtained, split into 12 theoretical dimensions related to food selection motives.

### 3.4. Construct Validation

A total of 788 subjects were included in the study (65% from the students and workers project and 35% from the second project); their descriptive characteristics are shown in Table 2. Over half of the participants were women (69.8%). Most participants reported being single, divorced, or widowed (83.6%), and over half of the participants (51.3%) presented some degree of food insecurity. The most frequent category was mild food insecurity (28.8%). Males had a higher proportion of high socioeconomic status than women (*p* = 0.012). Additionally, men exhibited higher proportions of employment status compared to women (56.3 vs. 44.0%). The mean age was 24.7 (8.5) years with no difference according to sex, while mean BMI was 24.4 (4.5) kg/m^2^ with men having higher values than women (*p* < 0.001). According to BMI classification, a higher proportion of overweight/obesity was found in men (53.4 vs. 38.4%). In terms of quality of food consumption, only 30.2% of the participants were in the category of “Healthy Food Intake”, and the most prevalent category was “Unhealthy Food Intake” (44.5%), without differences according to sex.

#### 3.4.1. Validity of the Updated-Food Choice Questionnaire

The EFA revealed that the best factor model (confirmed by CFA) included eight factors or dimensions (eigenvalues > 1), out of the 12 theoretical dimensions proposed in the content validation phase, which explained 54.6% of the variance. Standardized factor loadings for all items of these eight dimensions ranged from 0.34 to 0.82, showing a robust association between the items and their respective factors (see Table 3). The KMO for the data matrix was 0.96, indicating that the sample was adequate for factor analysis, while the Bartlett’s test was significant (x^2^ = 35,503.903, *p* < 0.001), revealing that the matrix had sufficient correlations.

The following eight factors were extracted (Table 3): (1) **health and** natural content (27 items), that covers aspects related to the impact of food on the body as well as the nutritional characteristics of food items; (2) **environmental and wildlife awareness** (12 items) reflects attitudes toward the environmental effects of food production and transportation, as well as the ethical treatment of animals; (3) **image management** (8 items) covers the affinity for known foods and the importance given to the image they project to other people; (4) **sensory appeal** (6 items) regards hedonic elements of eating, such as taste, smell, texture, and pleasure derived from food consumption; (5) **price** (5 items) relates to the economic aspects of food; (6) **convenience** (6 items) covers details of food preparation, cooking, and consumption; (7) **mood** (5 items) relates to the effects of food on the emotional state; and (8) **food identity** (6 items) assesses the use of food as a means of social interaction and the importance of food reflecting the individual’s culture.

The item “Has a long shelf life (doesn’t expire quickly)” did not meet the minimum required factor loading (0.30) and was subsequently removed from the questionnaire. As a result, a final 75-item questionnaire was obtained. The final 75-item U-FCQ is presented in Appendix A in English and Appendix A in Spanish.

When performing the CFA, the eight-structure model showed a significantly better fit and coherence among the items of its dimensions than the seven-factor and nine-factor models (Table 4).

Correlations between dimensions of the U-FCQ are shown in Table 5. There were moderate correlations (ranging from 0.30 to 0.73) between all the dimensions at the 0.05 significance level, except between **price** and **environmental and wildlife awareness** (0.26).

When we compared the AVE of each dimension with the SV of the correlation coefficients, we found that by using this method we were only able to establish discriminant validity in the following dimensions: **sensory appeal**, **convenience**, **price**, and **image management**. While using the cut-off point of 0.8 for the correlation coefficients between factors, we managed to obtain discriminant validity (coefficients ranged from 0.09 to 0.53) (Appendix A).

#### 3.4.2. Reliability of the Updated-Food Choice Questionnaire

The Cronbach’s alpha value for the complete tool was 0.97 and ranged from 0.74 to 0.96 for the dimensions (Table 3). According to the interpretation of their consistency, two dimensions were classified as excellent (**health and nutritional content** and **environmental and wildlife awareness**) and the remaining six as good [41].

For the ICC assessment, the U-FCQ was completed by 55 participants an average of 6.7 (standard deviation 3.4) months after the first application, with a minimum and maximum response period of 1 and 12 months, respectively. The ICC values for subscales of the U-FCQ ranged from 0.51 to 0.78; moreover, regarding its reliability, six dimensions were classified as moderate and two dimensions as good (**mood** and **environmental and wildlife awareness**) [42]. More information about the ICCs of each dimension is shown in Table 6.

### 3.5. Food Choice Motives Description

U-FCQ dimensions’ mean (SD) importance scores for the whole sample in the first application (test) are shown in Table 7. The most relevant food choice motives were: **sensory appeal**, **mood**, **health and nutritional content**, and **convenience**. On the other hand, the least important factors were: **image management**, **environmental and wildlife awareness**, and **food identity**. With the exception of **price** and **image management**, women significantly placed higher importance on all dimensions compared to men (*p* < 0.05).

## 4. Discussion

This study was the first effort to develop an updated, culturally adapted, and validated instrument, i.e., a 75-item U-FCQ, to measure traditional and emerging food choice determinants among Spanish-speaking adults.

We performed a rigorous methodological approach with several steps to conduct a validation study for the U-FCQ, giving a novel approach and significantly enhancing its associated robustness and reliability. First, we assessed the face validity of the original FCQ, considering, for each item, its relevance, clarity, representativeness and specificity. In addition, we performed a literature search to review previous adaptations and to identify dimensions or items that could be included in the tool. Although it has been suggested that a literature review should be the first step in the adaptation of a tool [23], we considered it important to explore how the original items performed, not only in the Spanish language of Mexico, but also in the sociocultural context of this country. By doing so, we identified the need for additional items suggested by the target population, as well as that the original structure of the tool was not directly applicable to our context, similar to what was found in several adaptations of this questionnaire [60,61,62]. Following the new items’ incorporation, we presented the updated questionnaire to a panel of NPs to obtain from them a quantitative and qualitative assessment, as well as suggestions for improvements. Next, we discussed the elimination or incorporation of new elements, prior to the construct validation phase, in which reliability assessment and factor analysis were performed. After following these steps, we obtained an eight-factor questionnaire (with 75 items), explaining 54.6% of the variance, with excellent to good internal consistency and good to moderate reproducibility.

In relation to the FCQ’s type of response, we opted to use the four-point Likert scale presented by the original version of the FCQ [8]. This differs from many of the previous adaptations, which have adopted a five-point scale [9,45] or even a seven-point scale [57], arguing that it allows for greater discrimination and the need to consider a neutral center point. However, four-point Likert scales have been described as being better for enabling the subject to form an opinion, preventing them from choosing a neutral point, particularly in the case of determining food selection motives [63].

Our study agrees with prior research reporting that the nine-factor model of the FCQ proposed by Steptoe and colleagues [8] may not be generalizable across different contexts, highlighting the need for culturally suitable instruments [64,65]. While our study identified similarities between the U-FCQ and the original FCQ [8], such as the retention of dimensions like **sensory appeal**, **convenience**, **mood**, and **price**, we also observed differences in their content due to higher factor loadings in alternative dimensions. This was the case for the **sensory appeal** and **price** dimensions, where the items “Is what I usually eat” and “Can be bought close to where I live or work” were added, respectively.

In the present study, the majority of items originally grouped under the **natural content** dimension exhibited higher loadings in the **health** dimension. Consequently, we named this dimension **health and natural content**. This convergence is similar to several FCQ adaptations [10,61,62,65,66,67,68], where the merging of these two original dimensions was reported. Additionally, items present in the **food policies/legislation** (proposed and generated during the face and content validation phase) were gathered under **health and natural content**. The higher number of items in the **health and natural content** dimension may be attributed to the fact that a significant proportion of the population were university students or professionals in the health area. Thus, it is possible that for these participants the nutritional characteristics of foods and aspects related to food labeling and adherence to nutritional policies and legislation are considered to a greater extent a general health aspect [69]. On the other hand, items from the original dimension **ethical concerns** and the dimension **sustainability**, that we proposed and generated in the face and content validation phase, were merged into the same factor, named **environmental and wildlife awareness**. This new dimension is consistent with similar constructs addressed in the FCQ adaptations from Lindeman et al. [51] and Verain et al. [57]. Finally, the proposed dimension **image management** remained after the factor analysis, however, other items were added (“Is like the food I ate when I was a child”, “Is in line with my religious beliefs”, and “Is gluten-free”).

In contrast to the work by Steptoe and collaborators, our study did not identify a specific dimension regarding weight control, as all items originally associated with this dimension were incorporated into the **health and natural content** dimension. A plausible explanation for this discrepancy is the demographic composition of our participants, a substantial portion of whom were university students in the health-related fields such as nutrition, medicine, nursing, among others. For these specific subjects, aspects related to weight control, like monitoring calories and fat content, could be perceived as a broader health-oriented food choice motive.

Concerning the fit of the model, we considered several indices. The RMSEA yielded satisfactory results, while the CFI and TLI produced values below the recommended threshold (0.9). The SMRS, however, showed acceptable scores. Concerning the above, it has been previously reported that the inconsistency between certain indices is not necessarily a diagnosis of problems in the model [70]. Rather, it is due to the fact that, by design, the indices evaluate fit from different perspectives [34,70,71]. For instance, it has been postulated that the CFI is more suitable for exploratory contexts, whereas the RMSEA is better aligned with confirmatory contexts [72]. Furthermore, we decided to retain the eight-factor model, as each dimension contained at least three items with internal logic among them.

When using the Lornell and Lacker criterion we were unable to establish discriminant validity for the dimensions **food identity**, **environmental and wildlife awareness**, **mood**, and **health and natural content**. This type of validity ensures that theoretically distinct constructs are indeed separate [37]; however, in this case, these dimensions may inherently overlap. For example, food identity may be closely related to health considerations, as individuals who prioritize certain ingredients may see this choice as an extension of their personal or cultural identity [73]. Similarly, environmental awareness may intertwine with broader health issues [6], making it difficult to fully differentiate these constructs. Additionally, this overlap could also result from participants’ perceptions, as they view these dimensions as interconnected aspects of their food choices. Nevertheless, the correlation coefficients between dimensions did not exceed 0.85, suggesting that there is not a strong relationship between them [40]. Future studies could compare how these dimensions perform in other contexts.

We observed moderate correlations between the extracted factors (except for **price** and **environmental and wildlife awareness**), in contrast to other studies that found high intercorrelations between dimensions [9,10,11,12]. Similarly to the original FCQ [8], we observed the highest associations between the **health and natural content** dimension and the rest of the food choice motives.

As for internal consistency, we obtained an excellent alpha for the complete tool and a stratified alpha from “excellent” to “good” for the dimensions. Through this assessment we were able to confirm that the dimensions identified after factor analysis contained items that in fact measured the same construct [41]. When analyzing the responses between the first and second application of the U-FCQ, a good to moderate reliability was observed. However, it should be noted that, although participants were invited to complete the tool a second time (15 days after their feedback), the period of time between test and retest was 6.7 months on average. In this sense, it is possible that subjects may have become more aware of their food choice behavior, meaning they could have either modified their behavior or given answers that they believed to be more acceptable [42].

Regarding the importance given to the dimensions extracted, **sensory appeal** was the predominant food choice motive among participants. This suggests that the hedonic aspect outweighs considerations of health and nutritional content [44]. These findings align with studies that have applied the FCQ in European countries [9], Turkey [60], Brazil [66], and Russia [48], as well as with literature reporting that the sensorial properties of food condition their predilection over other components [74]. Surprisingly, the dimension **mood** emerged as the second most important motive for food choices. This could be partly explained by the fact that more than half of the participants are women (69.8%), and it has been described that they are more prone to increased food consumption when they are under stress in comparison to men [75,76].

On the other hand, **image management** was found to be among the least important food choice motives. This may suggest that the studied population does not particularly care about previously known or consumed foods, as well as what the food they choose reflects about themselves. Moreover, this may be attributed, in part, to the relatively young age of the sample study (average of 24.7 years), who may have a greater desire to know and try new foods [77].

Similarly, the **environmental and wildlife awareness** dimension was given low importance when choosing food. In our questionnaire, this dimension includes items related to the effects of food production/transportation, as well as the ethical treatment of animals. From this perspective, it has been described that the individual’s lack of knowledge about concepts such as sustainable diets and animal rights can act as a barrier against embracing lifestyles that are more environmentally friendly [78]. Another plausible explanation is that the study population, as stated before, was predominantly comprised health science students and, therefore, could be more concerned about sufficient energy and nutrient intake than they are about the effects of food production systems on the environment [79].

This study presents several strengths. First, we proceed from the original version of the FCQ to determine the suitability of tool items, as well as to identify the need to update it. Previously, in 2016 and 2017, the FCQ was adapted for its application in the Mexican population; specifically, these studies aimed to identify food choice motives in Mexican consumers [13,14]. For this reason, our study differs from these two versions in several aspects, such as the study’s aim, dimensions considered, items per dimension, and criteria for their selection. The existing Mexican versions of the FCQ [13,14], were arbitrarily limited to three items per dimension, without specifying the rationale for this. In contrast, our adaptation has no limit of items per domain, since some of them measure complex constructs that would require a larger number of elements to be evaluated.

Second, we aimed to establish the processes to obtain an updated and culturally adapted FCQ and to evaluate not only construct validity but face and content validity as well. Although it is not a common practice to include the precise descriptions of face and content validity in questionnaire validation studies, they add further scientific value to enable replicating the methodology in other contexts. As a matter of fact, only the work of Diniz and collaborators [11], together with ours, delves into the methodology for assessing face and content validity. This agrees with the suggestions made by a systematic review on the use of the FCQ across cultures [7], where face and content validity are reported as important techniques in the methodology when adapting this questionnaire to different cultures and contexts. This allows the identification of items that may not be appropriate for the population targeted for application. Additionally, it is becoming increasingly common, and considered as a good practice, to present questionnaires used in the health field to focus groups to determine the quality of their items, in terms of their understandability [80].

Third, in this tool there is the expanded pool of items in comparison with other versions, which allows a more comprehensive evaluation of emerging food choice motives within the population. Our questionnaire achieves this by evaluating aspects that, to the best of our knowledge, have not been addressed in previous FCQ adaptations through the addition of new items developed by the research team, such as: “Does not cause me physical discomfort (allergies, stomach ache, diarrhea, etc.)”, “Helps me to meet my energy and nutritional needs”, “Is easy to digest”, “Is homemade”, “Keeps me full for a considerable period of time”, “Is low in salt”, “Has the least number of seals of warning” [58], “Sticks to the *Plato del Bien Comer* guide” [59], “Has an easy-to-understand nutrition labeling”, among others. Additionally, we incorporated the item “Is minimally processed” to evaluate the importance given to the avoidance of ultra-processed foods (UPFs), as well as the importance of food being as natural as possible. However, we did not use the UPF term, since in Mexico the population may not be aware of the degrees of food processing, thus it would not be clear for those unfamiliar with this concept [81]. Although the addition of new items could implicate an increase in response time, the estimated period for the 75-item tool application is approximately 15 to 20 min, which does not represent a significant burden for the subject. Additionally, its use can benefit the participant with personalized feedback to know the priority of the dimensions usually considered, allowing the identification of other important aspects to be considered in the decision-making process [7].

Finally, we present the complete version of the U-FCQ (Appendix A). Having the complete tool in both languages is important to compare translated versions from countries speaking the same language, allowing the assessment of the cultural effect. In addition, different meanings between the original language of the tool (English) and the language into which it is translated (Spanish) can be identified. In line with the above, when translating the questionnaire, we noticed that some items could have different meanings in Spanish than in English. For instance, two of the original Steptoe [8] items, “Takes no time to prepare” and “Is easy to prepare”, covered different meanings in the Spanish language, such as: (1) time of preparation, (2) ease of preparation, and (3) ease for consumption, therefore, we decided to separate them.

However, certain limitations must also be acknowledged. Despite including participants from the general population, we observed a higher response rate among health sciences professionals and students (65% of the sample). This demographic skew could potentially introduce bias, as individuals with a background in the health fields, due to the knowledge acquired during their academic training, may prioritize health considerations more strongly when selecting foods. Recognizing this is essential for interpreting findings in this study accurately. Another limitation is the absence of criterion-related validity for the U-FCQ, as there is no current “gold standard” or universally accepted method for assessing food choices [82]. This makes it challenging to validate this tool against an external benchmark. The latter highlights the need for further research to develop standardized methods for evaluating food-related behaviors.

On the other hand, due to the complexity and great variability of the food choice process, it is possible that certain items or aspects may be missing in this tool, which are necessary to fully understand this phenomenon. For example, because the U-FCQ primarily assesses the individual food choice motives, it does not consider factors external to the respondent that can significantly influence food choices (e.g., type of media to which the individual is exposed, peer pressure, climate, and so on). Moreover, our study did not account for food categories such as “superfoods” or dietary supplements, which are increasingly popular in some demographics and may play a role in food choice for health-conscious consumers [83]. Additionally, we did not assess specifically the importance of consumption of beverages (with or without alcohol), despite the fact that these can also significantly influence overall dietary patterns and food choices [84]. Future research could explore ways to integrate these external determinants and more specialized food items into the assessment to provide a more holistic understanding of food choice dynamics.

In addition, it should be noted that one of the key challenges in adapting and validating food-related questionnaires is addressing the cultural and contextual factors that influence food choice behavior [7]. In the case of the Mexican population, several aspects play a significant role in shaping food choices and purchasing decisions. Traditional dietary patterns, local food availability, socioeconomic factors, and family dynamics all contribute to how individuals prioritize certain food characteristics such as price, convenience, and sensory appeal [85]. Additionally, regional variations in food availability, particularly in rural versus urban areas, can impact responses in dimensions such as price or convenience [16]. Given these cultural nuances, it is important to consider that as the U-FCQ was validated within a specific population, the results and implications may not be fully generalizable to other cultural settings, even within the same country, without additional adjustments.

Furthermore, although in this work we propose a tool to be used in a Spanish-speaking population, it is essential to consider the cultural differences of each country, even when dealing with the same language. Thus, the U-FCQ can be used as a basis for the development of a specific and appropriate questionnaire for each context.

Moreover, it is recommended to perform the CFA in a different population than the one used for the EFA to avoid the “overfitting” bias [34]. However, due to constraints in sample size, it was not feasible to collect an additional sample for this purpose. Therefore, in this study both exploratory and factor analyses were conducted on the same sample. To mitigate this limitation, we applied strict criteria during the EFA to ensure that only well-supported factors were retained, thus minimizing the risk of identifying spurious factors. Future research should aim to replicate these findings in different populations to further confirm this structure.

Finally, this study has important implications for further research attempting to evaluate relevant food choice motives, as well as for the identification of the processes to face and content validate this tool. The U-FCQ could be used on a population level as a screening strategy and enable the development and follow-up of targeted intervention programs aimed at achieving a healthier food intake, food product innovations, and better food labeling and food marketing practices. Further studies employing complementary methodologies (biomarker assessments or genetic analysis) could provide valuable insights into the interplay between biological and psychosocial factors on food choices.

## 5. Conclusions

The current findings enabled the development of a clear, adequate, and valid U-FCQ composed of 75 items distributed in eight dimensions, which explain 54.6% of the variance, with an excellent to good Cronbach’s alpha and good to moderate ICC values. These outcomes support a satisfying validity and reliability with an effective culturally adapted tool for identifying food choice motives within a Spanish-speaking population from Mexico. In our sample, the individual and interpersonal factors (sensory appeal, mood, health and nutritional content, and convenience) were the primary motives for food choices. This study not only provides a useful instrument but also lays the groundwork for future cross-cultural comparisons and further adaptations of the FCQ in Latin American countries, expanding its applicability in wider settings. Ultimately, this work fills an important gap in the field, where culturally tailored and validated tools for studying food choices are crucial for understanding dietary behavior on a broader scale and their implications for public health and nutrition, providing a valuable resource for researchers and policymakers alike.

## Figures and Tables

**Table 1 nutrients-16-03749-t001:** Content validity assessment of the Updated-Food Choice Questionnaire items (72 items).

Theoretical Dimensions ^1^	Items in Spanish (as Validated) ^2^	Items in English (Proposed Translation) ^2^	Source	NP Disagreeing (n = 13)	CVR Value ^3^
Health (original)	*Sean ricos en vitaminas y minerales.*	Is rich in vitamins and minerals.	[8]	0	1
	*No me provoquen malestares físicos.*	Does not cause me physical discomfort.	New ^4^.	4	0.38 ^5^
	*Sean ricos en proteínas.*	Is rich in protein.	[8]	1	0.84
	*Sean buenos para mi piel, dientes, cabello, uñas,* etc.	Is good for my skin/teeth/hair/nails, etc.		0	1
	*Sean ricos en fibra.*	Is rich in fiber.		0	1
	*Me ayuden a cubrir mis necesidades energéticas y nutricionales.*	Helps me to meet my energy and nutritional needs.	New ^4^.	3	0.54
	*Sean nutritivos.*	Is nutritious.	[8]	1	0.84
	*Me mantengan saludable.*	Keeps me healthy.		0	1
	*Sean de fácil digestión.*	Is easy to digest.	New ^4^.	0	1
	*Me mantengan satisfecho(a) por un tiempo considerable.*	Keeps me full for a considerable period of time.		1	0.84
Weight control (original)	*Sean bajos en grasa.*	Is low in fat.	[8]	0	1
	*Me ayuden a controlar mi peso.*	Helps me to manage my weight.		1	0.84
	*Sean bajos en calorías (me permitan mantener un peso adecuado).*	Is low in calories (allows me to maintain an adequate weight).	[8] and modified by the research team.	1	0.84
Price (original)	*Sean baratos.*	Is cheap.	[8]	0	1
	*No sean costosos.*	Is not expensive.		1	0.84
	*Tengan una buena relación calidad-precio.*	Is good value for money.		0	1
Mood (original)	*Me ayuden a reducir el estrés.*	Helps me reduce stress.	[8]	3	0.54
	*Me hagan sentir bien (estado de ánimo).*	Makes me feel good (mood).		2	0.69
	*Me levanten el ánimo.*	Cheers me up.		0	1
	*Me ayuden a relajarme.*	Helps me relax.		1	0.84
	*Me ayuden a lidiar con la vida.*	Helps me cope with life.		6	0.07 ^5^
	*Me mantengan despierto(a) o alerta.*	Keeps me awake/alert.		1	0.84
Convenience (original)	*Puedan ser comprados cerca de donde vivo o trabajo.*	Can be bought close to where I live or work.	[8]	1	0.84
	*Sean prácticos (preparación fácil y consume casi inmediato).*	Is practical (easy preparation and almost immediate consumption).	New ^4^.	1	0.84
	*Puedan ser cocinados fácilmente.*	Can be easily cooked.	[8]	3	0.54
	*Estén disponibles en tiendas y supermercados.*	Is easily available in stores and supermarkets.		1	0.84
	*Tengan una vida de anaquel larga.*	Has a long shelf life.	[12]	4	0.38 ^5^
	*Puedan ser cocinados rápidamente*	Can be cooked quickly.	New ^4^.	3	0.54
Natural content (original)	*Sean bajos en hidratos de carbono.*	Is low in carbohydrates.	New ^4^.	4	0.38 ^5^
	*No contengan ingredientes artificiales (conservadores, colorantes, sustitutos/imitaciones de alimentos,* etc.).	Contains no artificial ingredients (preservatives, colorings, food substitutes/imitations, etc.).	[8] and modified by the research team.	2	0.69
	*No contengan aditivos (ingredientes añadidos, endulzantes, colorantes, conservadores, independientemente si son naturales o no).*	Contains no additives (added ingredients, sweeteners, colorants, preservatives, whether natural or not).		2	0.69
	*Sean mínimamente procesados.*	Is minimally processed.	New ^4^ and [52,55].	1	0.84
	*Sean libres de gluten.*	Is gluten-free.	New ^4^.	0	1
	*Contengan ingredientes naturales (en su mayoría).*	Contains mostly natural ingredients.	[8]	1	0.84
	*Sean bajos en grasa de origen animal.*	Is low in animal fat.	New ^4^.	2	0.69
	*No estén enlatados.*	Is not canned.		0	1
	*Sean bajos en azúcares.*	Is low in sugar.	New ^4^ and [46,49,53,54].	0	1
Familiarity (original)	*Sean de una marca conocida para mí.*	Is from a commercial brand known to me.	New ^4^.	1	0.84
	*Puedan ser consumidos en compañía de otras personas.*	Can be consumed in the company of other people.		3	0.54
	*Me sean familiares o conocidos.*	Is familiar or known to me.	[8]	1	0.84
	*Sean lo que usualmente consumo.* *Sean preparados en casa.*	Is what I usually eat. Is homemade.	New ^4^.	0	1
	*Sean parte de la dieta tradicional mexicana.*	Is part of the traditional Mexican diet.		0	1
	*Se parezcan a lo que consumía cuando era niño(a).*	Is like the food I ate when I was a child.	[8]	1	0.84
Sensory appeal (original)	*Tengan un sabor agradable.*	Tastes good.	[8]	0	1
	*Me sean placenteros.*	Is pleasant to me.	New ^4^.	1	0.84
	*Tengan un olor agradable.*	Smells nice.	[8]	0	1
	*Tengan un aspecto agradable.*	Looks nice.		0	1
	*Tengan una textura agradable.*	Has a pleasant texture.		0	1
Ethical concern (original)	*Provengan de países que apruebo políticamente.*	Comes from countries I approve of politically.	[8]	6	0.07 ^5^
	*Vayan acorde a mis creencias religiosas.*	Is in line with my religious beliefs.	[44,45,47,48,51]	1	0.84
	*Utilicen envases amigables con el medio ambiente.*	Is packaged in an environmentally friendly way.	[8]	0	1
	*Sean producidos sin explotación humana.*	Is produced without human exploitation.	[47,48,51,57]	6	0.07 ^5^
	*Mencionen claramente el país de origen.*	Has the country of origin clearly marked.	[8]	2	0.69
	*Sean producidos respetando los derechos de los animales.*	Is produced respecting animal rights.	[47,48,51,57]	0	1
	*Sean producidos sin que los animales hayan sufrido dolor.*	Is produced without animals having suffered pain.	[47,48,51]	3	0.54
Sustainability (proposed)	*Sean orgánicos.*	Is organic.	New ^4^ and [50,52].	2	0.69
	*No sean de origen animal.*	Is not of animal origin.	New ^4^.	2	0.69
	*Sean productos de temporada.*	Are seasonal products.	[57] and modified by the research team.	1	0.84
	*No contribuyan a la emisión de CO_2_.*	Does not contribute to CO_2_ emissions.	New ^4^.	3	0.54
	*No ocasionen daños al medio ambiente.*	Does not cause environmental damage.		2	0.69
	*No hayan sido transportados distancias excesivas.*	Has not been transported excessive distances.	[56,57]	6	0.07 ^7^
	*Sean productos locales y regionales.*	Are local and regional products.	[57]	0	1
	*Reduzcan la contaminación de suelo y agua.*	Reduces soil and water pollution.	[55]	2	0.69
Image management (proposed)	*Le den a las personas una buena impresión.*	Gives people a good impression.	[55]	6	0.07 ^5^
	*Tengan una publicidad atractiva.*	Has an attractive advertising.	New ^4^.	5	0.23 ^5^
	*Sean producidos por empresas reconocidas.*	Is produced by recognized companies.		2	0.69
	*Reflejen una imagen positiva de mí.*	Reflects a positive image of me.	[55]	2	0.69
	*Sean recomendados por profesionales de la salud.*	Is recommended by health professionals.	New ^4^.	0	1
Food policies/legislation (proposed)	*Contengan el menor número de sellos de advertencia.* [58] ^6^	Has the least number of seals of warning.	New ^4^.	1	0.84
	*Se apeguen al “Plato del Bien Comer”.* [59] ^7^	Sticks to the “Plato del Bien Comer” guide.		2	0.69
	*Tengan un etiquetado nutricional fácil de entender.*	Has an easy-to-understand nutrition labeling.		1	0.84

Abbreviations: NP: Nutrition professionals; CVR: Content validity ratio. ^1^ The original dimensions refer to those proposed by Steptoe and collaborators [8], while the proposed dimensions were formulated by the authors according to the nature of the items they contained prior to the factor analysis. ^2^ In English, all items are presented with the statement: “It is important to me that the food that I eat on a typical day…”, while in Spanish, they are introduced with the statement: “*Es importante para mí que los alimentos que consumo en un día común…*”. ^3^ CVR values range from 0 to 1, where 1 is absolute agreement. According to the number of NPs involved (n = 13), a minimum CVR of 0.54 was required to maintain an item in the questionnaire. ^4^ Items generated and proposed by the research team. ^5^ Items with a CVR lower than the minimum required (CVR < 0.54). ^6^ Mexico’s recent front-of-pack warning labels, which include 5 octagon-shaped seals that clearly, simply, and visibly indicate when a product contains an excess of nutrients and critical ingredients such as: calories, saturated fats, trans fats, sugar, and sodium. ^7^ Previous Mexican food guide for food promotion and health education, which establishes criteria for nutritional guidance. This food guide was updated in 2023, after this study’s performance.

**Table 2 nutrients-16-03749-t002:** Descriptive characteristics of participants of the construct validation phase.

Variable	Total (n = 788)	Women (n = 550)	Men (n = 238)	*p*-Value ^1^
Age, mean (SD)	24.7 (8.5)	24.8 (8.9)	24.5 (7.4)	0.629
Marital status, n (%)				
Single, divorced, or widowed	659 (83.6)	457 (83.1)	202 (84.9)	0.535
In a relationship ^2^	129 (16.4)	93 (16.9)	36 (15.1)	
Educational level, n (%)				
Basic level ^3^	537 (68.1)	384 (69.8)	153 (64.3)	0.126
Superior level ^4^	251 (31.9)	166 (30.2)	85 (35.7)	
Employment status, n (%)				
Unemployed	412 (52.3)	308 (56.0)	104 (43.7)	**0.002**
Employed	376 (47.7)	242 (44.0)	134 (56.3)	
SES ^5^, n (%)				
Low–middle	460 (58.4)	337 (61.3)	123 (51.7)	**0.012**
High	328 (41.6)	213 (38.7)	115 (48.3)	
BMI, mean (SD)	24.4 (4.5)	23.9 (4.5)	25.5 (4.7)	**<0.001**
BMI classification ^6^, n (%)				
Normal weight	448 (56.9)	339 (61.6)	111 (46.6)	**<0.001**
Overweight/obesity	340 (43.1)	211 (38.4)	127 (53.4)	
Food consumption quality ^7^, n (%)				
Healthy Food Intake	238 (30.2)	172 (31.3)	66 (27.7)	0.452
Habits in Need of Improvement	199 (25.3)	136 (24.7)	65 (27.3)	
Unhealthy Food Intake	351 (44.5)	242 (44.0)	107 (45.0)	
Food security status ^8^, n (%)				
Food security	384 (48.7)	267 (48.9)	117 (49.2)	0.761
Mild food insecurity	227 (28.8)	162 (29.5)	65 (27.3)	
Moderate food insecurity	109 (13.8)	72 (13.1)	37 (15.6)	
Severe insecurity	68 (8.6)	49 (8.9)	19 (8.0)	

Abbreviations: BMI: Body mass index; SD: Standard deviation; SES: Socioeconomic status. ^1^ Comparison of the mean of age and BMI between men and women was carried out using the *t*-test. Comparison of qualitative variables between sexes was performed using the chi-squared test. A value of *p* < 0.05 (shown in bold letters) was established as statistically significant for all analyses. ^2^ Married or common law. ^3^ Elementary, middle school, high school. ^4^ Bachelor’s degree, graduate degree, or medical specialty. ^5^ Assessed through the *Asociación Mexicana de Agencias de Inteligencia de Mercado y Opinión* (AMAI) tool [26]. The original 7 categories (A/B, C+, C, C−, D+, D, and E) were dichotomized into high (A/B and C+) and low–middle socioeconomic status (all the other categories). ^6^ Assessed through body mass index and dichotomized using the World Health Organization (WHO) categories [28]: normal (<25 kg/m^2^) and overweight/obese (≥25.0 kg/m^2^). ^7^ Categories obtained from the Mini-ECCA version 2 [30]. ^8^ Determined through the Latin American and Caribbean Scale of Food Security (its acronym in Spanish: ELCSA) [29].

**Table 3 nutrients-16-03749-t003:** Eigenvalues for the 8 extracted dimensions (explaining 54.6% of the variance), and factor loadings and reproducibility of the Updated-Food Choice Questionnaire items ^1^.

Dimensions (Eigenvalues) and Items ^2^	Factor Loading	Items’ Importance ^3^		Items’ Reproducibility ICC (95% CI)
		Test	Retest	
**Health and natural content (14.4)**				
56. Is nutritious.	0.75	3.4 (0.8)	3.8 (0.6)	0.53 (0.25, 0.64)
71. Keeps me healthy.	0.74	3.3 (0.9)	3.3 (0.9)	0.50 (0.22, 0.63)
57. Contains mostly natural ingredients.	0.73	3.1 (1.0)	3.3 (0.9)	0.57 (0.18, 0.60)
46. Is rich in vitamins and minerals.	0.71	3.3 (0.9)	3.6 (0.6)	0.70 (0.51, 0.89)
65. Is low in calories (allows me to maintain an adequate weight).	0.70	2.7 (1.0)	2.8 (1.0)	0.59 (0.26, 0.65)
28. Is low in fat.	0.70	2.8 (1.0)	3.1 (0.8)	0.51 (0.18, 0.61)
58. Is good for my skin/teeth/hair/nails, etc.	0.70	3.1 (1.0)	3.4 (0.9)	0.50 (0.20, 0.62)
60. Helps me to manage my weight.	0.70	3.0 (1.0)	3.2 (0.9)	0.44 (0.15, 0.60)
15. Helps me to meet my energy and nutritional needs.	0.69	3.3 (0.9)	3.6 (0.7)	0.57 (0.49, 0.89)
4. Is low in sugar.	0.69	2.7 (1.0)	3.2 (0.8)	0.60 (0.13, 0.58)
74. Contains no additives (added ingredients, sweeteners, colorants, preservatives, whether natural or not).	0.68	2.7 (1.0)	2.9 (0.9)	0.61 (0.38, 0.71)
31. Is rich in protein (e.g., eggs, beans, meat).	0.66	3.3 (0.8)	3.5 (0.7)	0.65 (0.31, 0.67)
36. Has the least number of seals of warning (i.e., excess sodium, calories, saturated fat).	0.65	2.8 (1.0)	3.2 (0.9)	0.56 (0.33, 0.69)
40. Sticks to the “*Plato del Bien Comer*” guide (food guide from Mexico).	0.64	2.8 (1.0)	3.1 (0.7)	0.72 (0.28, 0.66)
55. Contains no artificial ingredients (preservatives, colorings, food substitutes/imitations, etc.).	0.64	2.6 (1.0)	2.8 (1.1)	0.80 (0.69, 0.85)
22. Is recommended by health professionals.	0.64	2.8 (0.9)	3.1 (0.8	0.66 (0.35, 0.77)
13. Is rich in fiber.	0.64	2.7 (0.9)	3.1 (0.7)	0.52 (0.21, 0.79)
41. Is low in salt.	0.63	2.8 (0.9)	3.2 (0.8)	0.51 (0.24, 0.64)
63. Is minimally processed.	0.61	2.7 (1.0)	2.9 (1.0)	0.60 (0.35, 0.70)
39. Has an easy-to-understand nutrition labeling.	0.58	2.8 (1.0)	3.3 (0.7)	0.70 (0.51, 0.78)
70. Is low in carbohydrates.	0.58	2.4 (0.9)	2.6 (1.0)	0.60 (0.38, 0.71)
26. Is easy to digest.	0.56	2.9 (0.9)	3.3 (0.7)	0.45 (0.10, 0.58)
21. Is free of non-caloric sweeteners, for example: sucralose, stevia, etc.	0.55	2.4 (1.0)	2.6 (1.0)	0.52 (0.48, 0.79)
61. Is homemade.	0.55	3.2 (0.8)	3.2 (0.8)	0.47 (0.13, 0.59)
66. Is good value for money.	0.46	3.3 (0.8)	3.5 (0.7)	0.50 (0.15, 0.60)
6. Has the shortest storage time (as fresh as possible).	0.46	3.0 (0.9)	3.2 (0.8)	0.42 (0.11, 0.59)
48. Does not cause me physical discomfort (allergies, stomachache, diarrhea, etc.).	0.39	3.7 (0.6)	3.8 (0.4)	0.48 (0.15, 0.60)
**Environmental and wildlife awareness (7.7)**				
38. Is produced respecting animal rights.	0.78	2.4 (1.0)	2.6 (1.1)	0.34 (0.25, 0.46)
19. Is produced without animals having suffered pain.	0.76	2.3 (1.0)	2.5 (1.1)	0.51 (0.27, 0.66)
29. Reduces soil and water products.	0.74	2.6 (1.0)	2.7 (1.0)	0.52 (0.10, 0.57)
44. Does not cause environmental damage.	0.74	2.6 (1.0)	2.8 (1.0)	0.54 (0.29, 0.77)
69. Does not contribute to CO_2_ emissions.	0.71	2.5 (1.0)	2.5 (1.0)	0.50 (0.18, 0.61)
49. Is packaged in an environmentally friendly way.	0.64	2.8 (0.9)	2.9 (0.9)	0.61 (0.36, 0.70)
51. Is organic (free of fertilizers and pesticides).	0.56	2.4 (1.0)	2.4 (1.1)	0.59 (0.27, 0.66)
75. Is not of animal origin.	0.56	1.8 (0.9)	2.0 (1.0)	0.60 (0.33, 0.69)
33. Is low in animal fat.	0.51	1.9 (0.9)	2.3 (1.0)	0.42 (0.11, 0.58)
64. Has the country of origin clearly marked.	0.51	2.0 (0.9)	1.9 (1.0)	0.44 (0.06, 0.56)
50. Is not canned.	0.48	2.3 (0.9)	2.5 (1.0)	0.67 (0.49, 0.76)
10. Has not been transported long distances.	0.39	2.1 (0.9)	2.2 (1.0)	0.39 (0.10, 0.53)
**Sensory appeal (3.9)**				
25. Smells nice.	0.66	3.4 (0.7)	3.6 (0.6)	0.59 (0.34, 0.69)
5. Tastes good.	0.64	3.6 (0.7)	3.8 (0.5)	0.51 (0.24, 0.64)
30. Has a pleasant texture.	0.60	3.2 (0.8)	3.5 (0.6)	0.60 (0.36, 0.70)
45. Looks nice.	0.59	3.3 (0.8)	3.4 (0.8)	0.63 (0.40, 0.72)
59. Is pleasant to me.	0.50	3.4 (0.8)	3.5 (0.7)	0.68 (0.39, 0.71)
20. Is what I usually eat.	0.42	3.0 (0.8)	3.2 (0.7)	0.61 (0.37, 0.70)
**Image management (3.6)**				
73. Is from a commercial brand known to me.	0.68	1.7 (0.8)	1.5 (0.9)	0.45 (0.23, 0.57)
14. Has eye-catching advertising on their packaging.	0.68	1.5 (0.7)	1.7 (0.9)	0.63 (0.22, 0.62)
3. Is produced by companies I recognize from television, social media or others.	0.67	1.5 (0.7)	1.7 (0.8)	0.33 (0.13, 0.46)
52. Reflects a positive image of me.	0.48	1.9 (0.9)	2.0 (1.0)	0.71 (0.57, 0.80)
23. Is like the food I ate when I was a child.	0.47	1.8 (0.8)	2.0 (0.8)	0.51 (0.12, 0.58)
16. Is in line with my religious beliefs.	0.44	1.3 (0.7)	1.5 (0.9)	0.55 (0.23, 0.63)
1. Is gluten-free.	0.42	1.6 (0.9)	1.8 (1.0)	0.70 (0.53, 0.78)
8. Is considered good by other people.	0.41	2.2 (0.9)	2.0 (0.9)	0.54 (0.15, 0.60)
**Convenience (3.5)**				
47. Can be cooked quickly.	0.71	2.9 (0.9)	3.2 (0.8)	0.60 (0.25, 0.64)
72. Is practical (easy preparation and consumption).	0.68	3.1 (0.9)	3.3 (0.8)	0.57 (0.25, 0.65)
43. Can be consumed almost immediately.	0.63	2.4 (0.9)	2.4 (0.9)	0.53 (0.26, 0.65)
37. Can be easily cooked.	0.61	3.1 (0.8)	3.4 (0.8)	0.53 (0.18, 0.73)
17. Is familiar or known to me.	0.50	2.8 (0.9)	2.8 (0.9)	0.57 (0.10, 0.60)
53. Is easily available in stores, supermarkets, farmer’s markets, “*tianguis*”, etc.	0.34	3.3 (0.8)	3.3 (0.9)	0.55 (0.31, 0.68)
**Price (3.2)**				
18. Is cheap.	0.82	2.8 (0.8)	2.8 (0.7)	0.53 (0.28, 0.67)
12. Is not expensive.	0.78	3.0 (0.8)	3.0 (0.7)	0.51 (0.25, 0.65)
24. Is on sale.	0.67	2.2 (0.9)	2.4 (0.8)	0.69 (0.30, 0.88)
9. Keeps me full for a considerable period of time.	0.43	2.9 (0.9)	3.1 (0.9)	0.50 (0.26, 0.65)
11. Can be bought close to where I live or work.	0.39	3.2 (0.9)	3.1 (0.9)	0.50 (0.24, 0.64)
**Mood (2.7)**				
62. Cheers me up.	0.60	3.1 (0.9)	3.2 (0.9)	0.50 (0.18, 0.62)
67. Makes me feel good (mood).	0.56	3.1 (0.9)	3.2 (0.9)	0.51 (0.25, 0.72)
54. Helps me reduce stress.	0.53	2.7 (1.0)	2.8 (1.0)	0.33 (0.10, 0.52)
34. Keeps me awake/alert.	0.53	2.8 (0.9)	3.0 (0.9)	0.62 (0.26, 0.64)
32. Helps me get through life.	0.46	3.2 (0.9)	3.3 (0.8)	0.52 (0.22, 0.63)
**Food identity (2.5)**				
7. Are local and regional products.	0.48	2.3 (0.9)	2.3 (0.9)	0.56 (0.32, 0.69)
42. Can be consumed in the company of other people.	0.47	2.5 (1.0)	2.6 (1.0)	0.53 (0.07, 0.55)
27. Are seasonal products (typical of the season of the year, such as certain fruits and vegetables).	0.45	2.9 (1.0)	2.3 (0.9)	0.45 (0.10, 0.50)
68. Is produced in Mexico.	0.45	2.3 (0.9)	2.3 (0.9)	0.54 (0.29, 0.67)
35. Is consumed by most of the members of my household.	0.43	2.7 (1.1)	2.8 (1.1)	0.63 (0.43, 0.73)
2. Is part of the traditional Mexican diet.	0.40	1.9 (0.9)	2.5 (0.9)	0.50 (0.11, 0.67)

Abbreviations: SD: Standard deviation; ICC: Intra-class correlation; CI: Confidence interval. ^1^ Data are presented in decreasing order according to eigenvalues and factor loading. ^2^ In English, all items are presented with the statement: “It is important to me that the food that I eat on a typical day…”, while in Spanish, they are introduced with the statement: “*Es importante para mí que los alimentos que consumo en un día común*…”. ^3^ Data are presented as mean (standard deviation). Each item is answered on a 4-option Likert-type scale (1 = “not important at all” to 4 = “very important”), which is used to calculate the mean.

**Table 4 nutrients-16-03749-t004:** Confirmatory factor analysis fit indices of the Updated-Food Choice Questionnaire, according to number of factors in the model.

Model	X^2^	*df*	RMSEA (90% CI)	CFI	TLI	SRMR
7-factor (75 items)	10,126.72	2679	0.083 (0.079, 0.091)	0.711	0.702	0.072
8-factor (75 items)	10,084.94	2672	0.060 (0.059, 0.062)	0.779	0.770	0.071
9-factor (75 items)	10,296.45	2664	0.071 (0.069, 0.072)	0.757	0.748	0.070

Abbreviations: *df*: Degrees of freedom; RMSEA: Root mean squared error of approximation; CI: Confidence interval; CFI: Comparative fit index, TLI: Tucker–Lewis index; SRMR: Standardized root mean squared residual.

**Table 5 nutrients-16-03749-t005:** Correlation matrix of the dimensions of the Updated-Food Choice Questionnaire (n = 788).

	SA	M	HNC	C	P	FI	EWA
Mood (M)	0.57 *						
Health and natural content (HNC)	0.53 *	0.66 *					
Convenience (C)	0.55 *	0.55 *	0.50 *				
Price (P)	0.40 *	0.44 *	0.35 *	0.52 *			
Food identity (FI)	0.49 *	0.54 *	0.64 *	0.48 *	0.41 *		
Environmental and wildlife awareness (EWA)	0.40 *	0.54 *	0.73 *	0.40 *	0.26 *	0.62 *	
Image management	0.30 *	0.39 *	0.42 *	0.35 *	0.32 *	0.53 *	0.49 *

* *p* < 0.05.

**Table 6 nutrients-16-03749-t006:** Updated-Food Choice Questionnaire dimensions’ importance mean in test and retest, reproducibility, and internal consistency (n = 55).

U-FCQ Dimension	Dimension’s Importance ^1^		Reproducibility		Internal Consistency (CA) ^4^
	Test	Retest	ICC (CI 95%) ^2^	*R * ^3^ (*p*-Value)	
Sensory appeal	3.4 (0.5)	3.5 (0.7)	0.53 (0.18, 0.61)	0.42 (0.001)	0.80
Mood	3.0 (0.6)	3.1 (0.8)	0.78 (0.60, 0.81)	0.68 (<0.001)	0.84
Health and natural content	3.2 (0.7)	3.2 (0.5)	0.65 (0.28, 0.65)	0.56 (<0.001)	0.96
Convenience	3.1 (0.6)	3.0 (0.6)	0.51 (0.18, 0.61)	0.47 (<0.001)	0.80
Price	2.9 (0.6)	2.8 (0.5)	0.64 (0.45, 0.74)	0.61 (<0.001)	0.74
Food identity	2.7 (0.7)	2.6 (06)	0.69 (0.43, 0.73)	0.69 (<0.001)	0.79
Environmental and wildlife awareness	2.5 (0.7)	2.4 (0.7)	0.78 (0.60, 0.81)	0.80 (<0.001)	0.91
Image management	1.7 (0.6)	1.8 (0.5)	0.61 (0.37, 0.70)	0.74 (<0.001)	0.75

Abbreviations: U-FCQ: Updated-Food Choice Questionnaire; ICC: Intra-class correlation; CI: Confidence interval; CA: Stratified Cronbach’s alpha. ^1^ Data are presented as mean (standard deviation). ^2^ Reliability interpretation according to ICC values: poor (<0.50); moderate (0.50–0.75); good (>0.75–0.90); and excellent (>0.90) [42]. ^3^ Pearson’s correlation coefficient values. Interpretation of correlation magnitude: weak (≤0.39); moderate (0.40–0.69); strong (0.70–0.89); and very strong (0.90–1.00) [43]. ^4^ Obtained from the whole sample (n = 788).

**Table 7 nutrients-16-03749-t007:** Mean importance of Updated-Food Choice Questionnaire dimensions in descending order.

U-FCQ Dimension	Total (n = 788)	Women (n = 550)	Men (n = 238)	*p*-Value ^1^
Sensory appeal	3.3 (0.6)	3.4 (0.5)	3.1 (0.7)	<0.001
Mood	3.0 (0.7)	3.0 (0.7)	2.8 (0.8)	<0.001
Health and natural content	2.9 (0.7)	3.0 (0.6)	2.8 (0.7)	<0.001
Convenience	2.9 (0.6)	3.0 (0.6)	2.8 (0.7)	0.002
Price	2.8 (0.6)	2.8 (0.5)	2.8 (0.7)	0.215
Food identity	2.4 (0.7)	2.5 (0.7)	2.3 (0.7)	<0.001
Environmental and wildlife awareness	2.3 (0.7)	2.4 (0.7)	2.1 (0.7)	<0.001
Image management	1.7 (0.5)	1.7 (0.5)	1.6 (0.5)	0.167

Data are presented as mean (SD). Abbreviations: U-FCQ: Updated-Food Choice Questionnaire; SD: Standard deviation. ^1^ Comparison between the mean importance of the U-FCQ dimensions between men and women was carried out using the *t*-test for independent samples. A value of *p* < 0.05 was established as statistically significant.

## Data Availability

The datasets used and/or analyzed during the current study will be available from the corresponding author on reasonable request. The data are not publicly available due to the fact that the data are part of an ongoing study.

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
