# Peer review of "Updated-Food Choice Questionnaire: Cultural Adaptation and Validation in a Spanish-Speaking Population from Mexico"

_nutrients, 2024, doi:10.3390/nu16213749_

Round 1

Reviewer 1 Report (Previous Reviewer 3)

Comments and Suggestions for Authors

1. lines 271-272, “ intra-class correlation (ICC) was employed to assess test-retest reliability. The ICC was calculated for items and subscales scores of the two tests.” It was rare to calculate ICC for items, and no ICC for items was reported in the Result section. Besides, why ICC for total score of U-FCQ was not reported? Furthermore, could the authors also add Pearson's correlation coefficients for test-retest reliability?

2. For confirmatory factor analysis (CFA), in Table 4, did the authors make some model modification? Why the degrees of freedom were not correct as default models? Please report what kind of modification were executed in the manuscript.

3. For the new eight dimensions, some items in the dimension were not adequate, such as the Price dimension, the item 11 “Can be bought close to where I live or work.” seemed not to related to the price. Did the authors re-check the adequateness of each item for the its new dimension after exploratory factor analysis (EFA)? I suggest the authors be carefully check the items. Likewise, item 18 “Is cheap” and item 12 “Is not expensive” seemed too similar to each other. How about deleting one of them?

4. Strictly speaking, some criteria-related validity should be offered for U-FCD, the authors should listed the deficit as a limitation. Besides, for construct validity, the discrimination validation for the eight factors should also be tested, and average variance extracted (AVE) should be calculated to make sure the squared factor correlation did not beyond the AVE. If they will not reach the standard, take it as a limitation would be better.

5. Line 452, for ... an average 6.7 (3.4) months... the 3.4 was a standard deviation?

Author Response

Subject: Reply to comments of reviewers regarding the manuscript “Updated-Food Choice Questionnaire: Cultural Adaptation and Validation in a Spanish-speaking Population from Mexico”

Date: October 25th, 2024 

To the reviewers of Nutrients

Manuscript Nutrients-3279423

Respected reviewers, 

On behalf of the research team, we thank you for your valuable comments on our work. This review process is crucial to improve our manuscript. 

In the next lines, you fill find a point-by-point response at each one of your questions/observations. We hope that with these improvements our manuscript is now enough suitable for publication.

Kind regards,

Corresponding authors.

Reviewer #1

  1. Lines 271-272, “intra-class correlation (ICC) was employed to assess test-retest reliability. The ICC was calculated for items and subscales scores of the two tests.” It was rare to calculate ICC for items, and no ICC for items was reported in the Result section. Besides, why ICC for total score of U-FCQ was not reported? Furthermore, could the authors also add Pearson's correlation coefficients for test-retest reliability?

Response: Thank you for your suggestions. We have included the ICC for each item (Table 3), and we have added the Pearson correlation coefficients for test-retest reliability (Table 6). Regarding why we do not present the total ICC for the tool, it is important to note that the FCQ, both in its original version and this updated version, does not provide a total score, but rather each dimension’s importance scores. In this sense, this analysis can only be performed based on dimensions or factors.

  1. For confirmatory factor analysis (CFA), in Table 4, did the authors make some model modification? Why the degrees of freedom were not correct as default models? Please report what kind of modification were executed in the manuscript.

Response: We appreciate your feedback and the opportunity to clarify our methods. Regarding your question about model modifications in Table 4, we would like to confirm that we did not make any modifications to the CFA model after its initial specification. Following our exploratory factor analysis (EFA), we eliminated one item due to its low factor loading, resulting in a total of 75 items for all models tested. Each structure was rigorously evaluated to determine the most adequate fit for our data. We also acknowledge the oversight in reporting the degrees of freedom (df) in the original manuscript. We have corrected this error to accurately reflect the df in the final CFA results.

  1. For the new eight dimensions, some items in the dimension were not adequate, such as the “Price” dimension, the item 11 “Can be bought close to where I live or work.” seemed not to related to the price. Did the authors re-check the adequateness of each item for the its new dimension after exploratory factor analysis (EFA)? I suggest the authors be carefully check the items. Likewise, item 18 “Is cheap” and item 12 “Is not expensive” seemed too similar to each other. How about deleting one of them?

 Response: Thank you for your insightful feedback. Once we conducted the EFA, each extracted dimension was reviewed in depth to ensure that the items within a factor were coherent with among themselves. Regarding item 11, "Can be bought close to where I live or work," within the "Price" dimension, we kept this item under the "Price" dimension (as identified by the EFA) because proximity can indirectly influence perceived costs for consumers. When a product is available near a person's home or workplace, it reduces transportation-related expenses and time costs, which consumers often associate with the overall price of obtaining the product [1].

In the context of the Mexican population, this perception is particularly relevant, as convenience and accessibility to food outlets can play a crucial role in determining the affordability of certain products [2]. Consumers may view products available nearby as more cost-effective due to the reduced need for travel. Furthermore, we believe that this finding (where item 11 shows a higher factor loading in the "Price" dimension) could be influenced by the specific characteristics of our sample. Approximately 60% of the participants were university students, who may have different priorities and constraints compared to the general population. This group are more likely to rely on nearby food outlets due to their limited time and transportation options, thus reinforcing the association between geographical proximity and perceived price. For this demographic, convenience often equates to lower overall costs, as they may prioritize proximity to reduce time and expenses linked to accessing food.

As for the items "Is cheap" and "Is not expensive", we understand that these might appear similar at first glance, but we believe they capture different aspects of how consumers assess the economic cost of food, particularly in the Mexican context. Item 18, "Is cheap," directly refers to the perception of a product as being of low cost. It reflects the consumer’s evaluation of the product’s affordability in absolute terms. In contrast, item 12, "Is not expensive," portrays a more comparative evaluation [3]. This item is designed to capture whether the consumer perceives a product to be relatively affordable when compared to other available alternatives or to their expected price range. Essentially, it is less about the absolute low cost of the product and more about whether the product is priced fairly or below what the consumer might anticipate.

These two items were included to account for different consumer attitudes toward pricing. Some consumers might judge a product as "cheap" in absolute terms, while others may judge it as "not expensive" relative to their expectations or alternatives. By keeping both items, we can capture a more nuanced understanding of price perception and ensure the robustness of the "Price" dimension. We hope this explanation clarifies the rationale for retaining both items.

In addition to this, is not rare to see in validation tools that it is usual to keep some items may be seemed as repetitive, but that confirms subjects’ perception about a specific dimension [4–6], in this case, price.

References

  1. Hasanzade, V.; Elshiewy, O.; Toporowski, W. Is It Just the Distance? Consumer Preference for Geographical and Social Proximity of Food Production. Ecological Economics 2022, 200, 107533, doi:https://doi.org/10.1016/j.ecolecon.2022.107533.
  2. Krstikj, A.; Egurrola Hernández, E.A.; Giorgi, E.; Garnica Monroy, R. Evaluating the Availability, Accessibility, and Affordability of Fresh Food in Informal Food Environments in Five Mexican Cities. J Urban Aff 1–16, doi:10.1080/07352166.2023.2276778.
  3. Campbell, D.; Aravena, C.D.; Hutchinson, W.G. Cheap and Expensive Alternatives in Stated Choice Experiments: Are They Equally Considered by Respondents? Appl Econ Lett 2011, 18, 743–747, doi:10.1080/13504851.2010.498341.
  4. Einola, K.; Alvesson, M. Behind the Numbers: Questioning Questionnaires. Journal of Management Inquiry 2021, 30, 102–114, doi:10.1177/1056492620938139.
  5. Suárez-Alvarez, J.; Pedrosa, I.; Lozano, L.M.; García-Cueto, E.; Cuesta, M.; Muñiz, J. Using Reversed Items in Likert Scales: A Questionable Practice. Psicothema 2018, 30, 149–158, doi:10.7334/psicothema2018.33.
  6. Ward, M.K.; Meade, A.W. Annual Review of Psychology Dealing with Careless Responding in Survey Data: Prevention, Identification, and Recommended Best Practices. Annu. Rev. Psychol. 2023 2024, 74, 577–596, doi:10.1146/annurev-psych-040422.

  1. Strictly speaking, some criteria-related validity should be offered for U-FCQ, the authors should list the deficit as a limitation. Besides, for construct validity, the discrimination validation for the eight factors should also be tested, and average variance extracted (AVE) should be calculated to make sure the squared factor correlation did not beyond the AVE. If they will not reach the standard, take it as a limitation would be better.

Response: Thank you for your thoughtful feedback. We agree that, in an ideal scenario, criterion-related validity would strengthen the evaluation of the U-FCQ. However, as there is no universally accepted gold standard for assessing food choices, establishing criterion-related validity is not viable in this context. The lack of such a standard means that comparing the U-FCQ against an external criterion is not feasible. Nonetheless, we acknowledge this as a limitation in the current study and have included this point in the revised manuscript (lines 689-693).

For discriminant validity, we used two approaches, the Fornell and Larcker criterion and the assessment of the correlation coefficient values between factors. We discuss our results in lines 575-587, and the results tables are presented in Supplementary Table 4.

  1. Line 452, for “... an average 6.7 (3.4) months...” the 3.4 was a standard deviation?

Response: Thank you for your observation. Yes, the 3.4 corresponds to the standard deviation. We have included the specification in line 468.

Reviewer 2 Report (Previous Reviewer 1)

Comments and Suggestions for Authors

The article titled "nutrients-3279423_ Updated-Food Choice Questionnaire: Cultural Adaptation and Validation in a Spanish-speaking Population", sets out to: 1) assess the face validity (FV) of the original 36-item FCQ, 2) develop an Updated-FCQ (U-FCQ) and evaluate its content validity (CV) (instrument suitability), and 3) assess its construct validity and reliability in a Spanish-speaking population. This paper is submitted to the "Nutrition and Public Health" section.

The title is informative and reflects the content of the study; however, it should specify that the research focuses on the Mexican population.

The abstract provides a concise summary of the study, covering all the key sections. It should be made clear that the study involves the Mexican population.

The keywords should be reviewed and adjusted in line with MeSH classification.

The introduction discusses recent changes in lifestyle habits, highlighting the need to update food consumption surveys, particularly in relation to dietary changes in Mexico. This context justifies the study's objective, which is clearly structured into three parts, as outlined.

In the methods section, a cross-sectional design is presented, incorporating exploratory and confirmatory analyses on Mexican participants. The study is based on the original 36-item FCQ applied to the Mexican population, assessing its suitability in the current context. The tool was translated into Spanish (for the Mexican context) using a translation and back-translation method, conducted by the first author, who is fluent in both Spanish and English.

A literature review was conducted up to December 2021, and content validation was performed at the University of Guadalajara. Construct validation of the U-FCQ was carried out through a cross-sectional validation study involving Mexican adults aged 18 and older, using data from two separate research projects. The study must also mentions approval from the ethics committee, which authorised the research.

The statistical analysis used is standard for questionnaire validation.

Results: The results are presented in a structured manner, making the study easy to follow. The tables summarise the findings concisely, aiding comprehension. Overall, the results are clearly presented.

Discussion: It should be noted that food-related questionnaires face significant challenges depending on the culture and population being studied. These cultural aspects should be considered in the discussion.

The discussion also addresses the key findings and relates them to the existing literature. It recognises the complexity of these analyses due to the variability in food choices, which may result in certain foods being overlooked. This should be clearly stated as a limitation in the discussion section.

The conclusion should focus on the contribution of the study to the field, rather than simply summarising the work. I believe the conclusion needs to be rewritten.

Overall, this is a strong research study, although it requires some clarifications as I have pointed out. Most importantly, the title should specify that the study involves the Mexican population, as indicated in the conclusion. This should also be reflected in the objectives, as the Spanish-speaking population is broad, covering many countries where food culture and dietary habits differ significantly.

Author Response

Subject: Reply to comments of reviewers regarding the manuscript “Updated-Food Choice Questionnaire: Cultural Adaptation and Validation in a Spanish-speaking Population from Mexico”

Date: October 26th, 2024 

To the reviewers of Nutrients

Manuscript Nutrients-3279423

Respected reviewers, 

On behalf of the research team, we thank you for your valuable comments on our work. This review process is crucial to improve our manuscript. 

In the next lines, you fill find a point-by-point response at each one of your questions/observations. We hope that with these improvements our manuscript is now enough suitable for publication.

Kind regards,

Corresponding authors.

Reviewer #2

  1. The title is informative and reflects the content of the study; however, it should specify that the research focuses on the Mexican population.

Response: Thank you for your observation. We agree with you that it would be adequate to specify that the sample focuses on the Mexican population. In response to this, we have included this in the title, abstract (line 37) introduction (line 107), and conclusion (line 745), keeping also the specification that this is a Spanish-speaking population.

  1. The abstract provides a concise summary of the study, covering all the key sections. It should be made clear that the study involves the Mexican population.

Response: Thank you for your suggestion. This has been added in line 37.

  1. The keywords should be reviewed and adjusted in line with MeSH classification.

Response: Thank you for your insight. The keywords have been adjusted to fit MeSH terms. However, we kept the term “Food Choice Questionnaire” which, although it is not a MeSH term, we consider important to be included for the purposes of this study.

  1. Construct validation of the U-FCQ was carried out through a cross-sectional validation study involving Mexican adults aged 18 and older, using data from two separate research projects. The study must also mention approval from the ethics committee, which authorized the research.

Response: Thank you for your input. This information is in lines 174 and 175 and in the “Institutional Review Board Statement” section (lines 778-781).  In addition, we have indicated which code of approval by the ethics committees corresponds to each project (line 780).

  1. Discussion: It should be noted that food-related questionnaires face significant challenges depending on the culture and population being studied. These cultural aspects should be considered in the discussion.

Response: Thank you for your valuable comment. We fully agree that cultural and population-specific factors can significantly influence how individuals perceive and prioritize different aspects of food choice. Therefore, we have included in the discussion that for a more accurate use and interpretation of food-related questionnaires, researchers should take into account these cultural nuances by considering local dietary habits, food availability, and societal attitudes towards food. Also, we recognize that the items and dimensions of the U-FCQ may resonate differently in various cultural settings, even within the same country, due to factors like family food traditions, economic constraints, and local food preferences (lines 708-719).

  1. The discussion also addresses the key findings and relates them to the existing literature. It recognizes the complexity of these analyses due to the variability in food choices, which may result in certain foods being overlooked. This should be clearly stated as a limitation in the discussion section.

Response: Thank you for this interesting observation. It is true that due to the complexity of the food selection process, different aspects were overlooked. In response to this, although we briefly mentioned this in lines 694-699, we have included in the discussion some other aspects that our tool does not evaluate (lines 699-706).

  1. The conclusion should focus on the contribution of the study to the field, rather than simply summarizing the work. I believe the conclusion needs to be rewritten.

Response: Thank you for your suggestion. As a response, we have modified the conclusion to provide more emphasis on the contributions of this study to the field (lines 747-754).

  1. Overall, this is a strong research study, although it requires some clarifications as I have pointed out. Most importantly, the title should specify that the study involves the Mexican population, as indicated in the conclusion. This should also be reflected in the objectives, as the Spanish-speaking population is broad, covering many countries where food culture and dietary habits differ significantly.

Response: Thank you for your feedback. We have addressed your comments by specifying that the study focuses on the Mexican population, in the title, abstract (line 37) introduction (line 107), and conclusion (line 745). Additionally, we have kept that it is a Spanish-speaking population because not all inhabitants of Mexico speak Spanish.

Round 2

Reviewer 1 Report (Previous Reviewer 3)

Comments and Suggestions for Authors

OK

This manuscript is a resubmission of an earlier submission. The following is a list of the peer review reports and author responses from that submission.

Round 1

Reviewer 1 Report

Comments and Suggestions for Authors

Thank you very much for allowing me to review the article titled " Updated-Food Choice Questionnaire: Cultural Adaptation and Validation in a Spanish-speaking Population " (nutrients-3139039), The objectives of this study were: 1) to evaluate the face validity of the FCQ, 2) to update the FCQ (U-FCQ) and to assess its content validity, and 3) to assess the construct validity and reliability of the U-FCQ in a Spanish-speaking population.

The introduction clearly outlines the complexity involved in food choice and the various factors from different domains that influence it. It also discusses the different instruments developed to capture the motivations behind food choice, specifically introducing the Food Choice Questionnaire (FCQ), developed in 1995 by Steptoe and collaborators. The FCQ has been translated and adapted to various contexts and populations, with more than 30 modified versions currently available, most of which are in English, although it has also been applied in other contexts such as Mexico. Since there are no studies evaluating the content validity of these versions, this study was undertaken. However, the introduction should address the differences between various cultural contexts, including those within Spanish-speaking populations, especially regarding food intake, which has distinct idiosyncrasies and cultural variations across different countries. It is crucial to justify the inclusion of a single questionnaire for Spanish-speaking populations, given the significant dietary and nutritional heterogeneity rooted in diverse cultures and food availability.

In line 158, it is mentioned that the study was conducted on Mexicans over 18 years old, but this should have been structured earlier. In line 170, the sample size calculation is discussed, but the method for calculating the 760 participants is not explained. In line 171, it is stated that participants were recruited directly, but the recruitment process is not described. I believe this section should be better organised.

Regarding the materials and methods section, a mixed design based on a cross-sectional study design is applied. It should specify the population in which this design was implemented, the inclusion and exclusion criteria, the sample size, and whether the sample size was calculated for this validation. If the study involved both Mexican and Spanish populations, the proportion of the sample from each region and the procedure followed should be clarified. This is a key element for consolidating the results, and any differences between the populations should be mentioned.

In section 2.3, it is stated that a literature review was conducted from August to December 2021. This review should extend until 2024. The methodology is clearly and adequately structured for the stated purpose.

Regarding the results, section 3.1 discusses the evaluation panel, consisting of nutrition degree students and three advanced science students who evaluated the questionnaires administered to the Mexican population. This information should actually be included in the methodology section.

In the results section, the instrument's validation is presented based on the 788 subjects who participated, whereas the materials and methods section calculated a sample size of 760. The number of individuals who agreed to participate and the response rate should be indicated. The tables are illustrative and correctly present the information, and the interpretation of the results aligns with the obtained data.

In the discussion, the validity of the studied questionnaire should be considered, as it pertains to a Mexican population, which, although Spanish-speaking, may not be applicable to other Spanish-speaking countries due to cultural differences. This is a fundamental limitation that should be addressed in the conclusions as well.

Author Response

Subject: Reply to comments of reviewers regarding the manuscript “Updated-Food Choice Questionnaire: Cultural Adaptation and Validation in a Spanish-speaking Population”

Date: October 10th, 2024 

To the reviewers of Nutrients

Manuscript Nutrients-3139039

Respected reviewers, 

On behalf of the research team, we thank you for your valuable comments on our work. Undoubtedly, the review process is vital to improve our work and continue to grow as researchers. 

Although the paper was rejected, below you will find answers to each of your suggestions and comments, hoping that it can be assessed and still be considered for publication in Nutrients.

Moreover, we would like to clarify why we are presenting the manuscript as it is. We originally had two different manuscripts: one for the face and content validation phase (which included part of the updating process of the Food Choice Questionnaire (FCQ) (nutrients-2963281), and the other one for the construct validation phase. In fact, we presented the face and content validation manuscript for peer review process (nutrients-2963281), and reviewers “did not see the scientific value of publishing a paper in which the instrument is not fully validated”, and suggested to “publish a unique paper that demonstrates the full validity of the proposed new instrument for testing food choice”. For this reason, we prepared a single manuscript with face, content, and construct validity of an Updated-Food Choice Questionnaire.

We hope that this could help to better understand the present manuscript, attached as Word file, and to consider it for publication.

Best regards,

Corresponding authors.

Reviewer #1

  1. The introduction should address the differences between various cultural contexts, including those within Spanish-speaking populations, especially regarding food intake, which has distinct idiosyncrasies and cultural variations across different countries.

Thank you for your suggestion. Although we had briefly described this on lines 87-89 and 98-100, we added that this could happen too in Spanish-speaking populations on lines 100-101.

  1. It is crucial to justify the inclusion of a single questionnaire for Spanish-speaking populations, given the significant dietary and nutritional heterogeneity rooted in diverse cultures and food availability.

Thank you for your comment. Actually, we do not propose this updated questionnaire as a single questionnaire for Spanish-speaking populations. We highlighted the “Spanish-speaking population” since items were translated to Spanish and validated in that language; therefore, they are different from those originally proposed by Steptoe (which are in English), and for that reason, we added a Supplementary material (Appendix B) where you can find the both, the Spanish and the proposed English translation for the tool. Also, in the discussion we included some explanation about the necessity of validating the tool in different contexts, including Spanish-speaking populations (lines 661-664). 

  1. In line 158, it is mentioned that the study was conducted on Mexicans over 18 years old, but this should have been structured earlier.

Thank you for your comment. However, since we are describing the methodology for the construct validation phase (which is different from the face and content validation phase), and this is mentioned in the first lines of that phase, we consider that we could leave the participants age in the same place (line 168).

  1. In line 170, the sample size calculation is discussed, but the method for calculating the 760 participants is not explained.

Thank you for your comment. We described with more detail the sample size calculation on lines 177-181.

  1. In line 171, it is stated that participants were recruited directly, but the recruitment process is not described. I believe this section should be better organized.

Thank you for your suggestion. We now include this information on lines 182-189.

  1. Regarding the materials and methods section, a mixed design based on a cross-sectional study design is applied. It should specify the population in which this design was implemented, the inclusion and exclusion criteria, the sample size, and whether the sample size was calculated for this validation.

Thank you for your suggestion. We now include information about design (lines 167-168), inclusion and exclusion criteria (lines 190-199) and sample size and its calculation (lines 177-181).

  1. If the study involved both Mexican and Spanish populations, the proportion of the sample from each region and the procedure followed should be clarified. This is a key element for consolidating the results, and any differences between the populations should be mentioned.

The study was performed only in Mexican participants. This has been specified on lines 115 and 168.

  1. In section 2.3, it is stated that a literature review was conducted from August to December 2021. This review should extend until 2024. The methodology is clearly and adequately structured for the stated purpose.

Thank you for your comment. It is important to notice that we are carrying out a systematic review and, for that review, we are including papers published until 2024. However, the review for updating the tool was performed before the construct validation phase, and items were validated according to what we had identified until 2021. For this reason, we decided to keep the dates for the literature review as they are.

  1. Regarding the results, section 3.1 discusses the evaluation panel, consisting of nutrition degree students and three advanced science students who evaluated the questionnaires administered to the Mexican population. This information should actually be included in the methodology section.

Thank you for your suggestion. We consider that the description of the nutritional professionals accepting to participate in the validation is part of the results of the study; for this reason, we decided to keep this information in the results section.

  1. In the results section, the instrument's validation is presented based on the 788 subjects who participated, whereas the materials and methods section calculated a sample size of 760.

Thank you for expressing your doubt. The sample size of 760 is the minimum expected by means of including at least 10 participants per item present in the tool to be validated (lines 177-181). However, at the end, 788 subjects agreed to participate, and we considered them all.

  1. The number of individuals who agreed to participate and the response rate should be indicated.

Thank you for your suggestion. This has been added on line 372.

  1. In the discussion, the validity of the studied questionnaire should be considered, as it pertains to a Mexican population, which, although Spanish-speaking, may not be applicable to other Spanish-speaking countries due to cultural differences. This is a fundamental limitation that should be addressed in the conclusions as well.

Thank you for your suggestion. This has been added on lines 661-664.

Reviewer 2 Report

Comments and Suggestions for Authors

Dear authors

You have put all your research work (at least two distingtive themes: 1) assesing the validity of FCQ, 2) updating the FCQ) in one manuscript! The presentation of two subject matters in one text makes the manuscript unclear, the paragraphs in an upside order.

I recommend to have the research work and data of the updating the FCQ rewritten in one manuscript and resubmitted for evaluation.   

Author Response

Subject: Reply to comments of reviewers regarding the manuscript “Updated-Food Choice Questionnaire: Cultural Adaptation and Validation in a Spanish-speaking Population”

Date: October 10th, 2024 

To the reviewers of Nutrients

Manuscript Nutrients-3139039

Respected reviewers, 

On behalf of the research team, we thank you for your valuable comments on our work. Undoubtedly, the review process is vital to improve our work and continue to grow as researchers. 

Although the paper was rejected, below you will find answers to each of your suggestions and comments, hoping that it can be assessed and still be considered for publication in Nutrients.

Moreover, we would like to clarify why we are presenting the manuscript as it is. We originally had two different manuscripts: one for the face and content validation phase (which included part of the updating process of the Food Choice Questionnaire (FCQ) (nutrients-2963281), and the other one for the construct validation phase. In fact, we presented the face and content validation manuscript for peer review process (nutrients-2963281), and reviewers “did not see the scientific value of publishing a paper in which the instrument is not fully validated”, and suggested to “publish a unique paper that demonstrates the full validity of the proposed new instrument for testing food choice”. For this reason, we prepared a single manuscript with face, content, and construct validity of an Updated-Food Choice Questionnaire.

We hope that this could help to better understand the present manuscript, attached as a Word file, and to consider it for publication.

Best regards,

Corresponding authors.

Reviewer #2

  1. You have put all your research work (at least two distinctive themes: 1) assessing the validity of FCQ, 2) updating the FCQ) in one manuscript! The presentation of two subject matters in one text makes the manuscript unclear, the paragraphs in an upside order. I recommend to have the research work and data of the updating the FCQ rewritten in one manuscript and resubmitted for evaluation.   

 Thank you for your suggestion. However, we previously had this manuscript divided into two parts, but in the previous Nutrients peer review phase (manuscript nutrients-2963281), reviewers considered that we had to include all this information (the updating process and the tool’s validity) in one single manuscript. For this reason, and since that manuscript was rejected, we sent our manuscript as a new one.

Reviewer 3 Report

Comments and Suggestions for Authors

1. Since the new “Food Choice Questionnaire (FCQ)“ has very different dimensions (factors) and items, why did the authors not give it a new name rather than an updated one (U-FCQ)? I think it is better to rename the new questionnaire.

2. For construct validity, why EFA and CFA use the same sample? It would be better to collect a new sample to confirm the construct validity by CFA. Maybe it needs to be taken as a limitation.

3. For CFA, many fit indexes should be reported, such as chi-squared value, degrees of freedom, GFI, AGFI, CFI, SRMR, and RMSEA. In general, if the fit-indexes were not acceptable, the construct validity can not be satisfied. Besides, did the authors use the default setting for factor correlation in CFA? What are the correlation coefficients among the factors in the CFA? They involve discrimination validity and convergent validity.

4. The new questionnaire (U-FCQ) was in lack of criterion-related validity. Can the authors offer the reason?

5. In Table 4, for the test-retest reliability, why did the authors use ICC rather than Pearson's correlation coefficient? Besides, which ICC was used that should be reported? ICC(2,1)?

6. line 317, the denotation number was a typo? Was it “7” rather than “3”?

7. line 369, the abbreviation “FI” should be defined clearly.

8. line 401, the title of the Table 3 “…factor loadings for the of the…” should be deleted either “for the” or “of the.”

Author Response

Subject: Reply to comments of reviewers regarding the manuscript “Updated-Food Choice Questionnaire: Cultural Adaptation and Validation in a Spanish-speaking Population”

Date: October 10th, 2024 

To the reviewers of Nutrients

Manuscript Nutrients-3139039

Respected reviewers, 

On behalf of the research team, we thank you for your valuable comments on our work. Undoubtedly, the review process is vital to improve our work and continue to grow as researchers. 

Although the paper was rejected, below you will find answers to each of your suggestions and comments, hoping that it can be assessed and still be considered for publication in Nutrients.

Moreover, we would like to clarify why we are presenting the manuscript as it is. We originally had two different manuscripts: one for the face and content validation phase (which included part of the updating process of the Food Choice Questionnaire (FCQ) (nutrients-2963281), and the other one for the construct validation phase. In fact, we presented the face and content validation manuscript for peer review process (nutrients-2963281), and reviewers “did not see the scientific value of publishing a paper in which the instrument is not fully validated”, and suggested to “publish a unique paper that demonstrates the full validity of the proposed new instrument for testing food choice”. For this reason, we prepared a single manuscript with face, content, and construct validity of an Updated-Food Choice Questionnaire.

We hope that this could help to better understand the present manuscript, attached as a Word file, and to consider it for publication.

Best regards,

Corresponding authors.

Reviewer #3

  1. Since the new “Food Choice Questionnaire (FCQ)” has very different dimensions (factors) and items, why did the authors not give it a new name rather than an updated one (U-FCQ)? I think it is better to rename the new questionnaire.

Thank you for your suggestion. We think that it is very reasonable; however, since we started from the original Food Choice Questionnaire, we consider that it could be unethical to leave out the original name. For this reason, we named this questionnaire as Updated Food Choice Questionnaire.

  1. For construct validity, why EFA and CFA use the same sample? It would be better to collect a new sample to confirm the construct validity by CFA. Maybe it needs to be taken as a limitation.

Thank you for your observation. We agree with you that ideally the EFA and CFA should be applied in different populations. We have established this as a limitation in our study (lines: 665-668). Additionally, we mention the ways in which we tried to mitigate this (lines: 668-672).

  1. For CFA, many fit indexes should be reported, such as chi-squared value, degrees of freedom, GFI, AGFI, CFI, SRMR, and RMSEA. In general, if the fit-indexes were not acceptable, the construct validity can not be satisfied. Besides, did the authors use the default setting for factor correlation in CFA? What are the correlation coefficients among the factors in the CFA? They involve discrimination validity and convergent validity.

Thank you for your observation. We now include information about chi-square, degrees of freedom, RMSEA (90% CI), CFI, TLI, and SRMR, and the correlation coefficients between the U-FCQ dimensions, in the methods section (lines 260-266), the results section (lines 430-445) and the discussion (lines 542-555). Also, we include these results for the 7 and 9-factor model, so we are not using the default setting for factor correlation in CFA (Table 4, lines 433-437).

  1. The new questionnaire (U-FCQ) was in lack of criterion-related validity. Can the authors offer the reason?

The reason for this is that there is not a gold standard for perceived food choice motives. Because of this, we report face, content and construct validity, besides reproducibility.

  1. In Table 4, for the test-retest reliability, why did the authors use ICC rather than Pearson's correlation coefficient? Besides, which ICC was used that should be reported? ICC(2,1)?

Thank you for your observation. Although Person’s correlation coefficient can be reported, this analysis measures association between two variables, but not the agreement and consistency between them. In other words, it can be possible that variables with poor agreement are highly correlated, which is not desirable in order to assess reproducibility. The ICC reported is the one called as “individual” in STATA, which measures agreement (Martínez-González MA, Sánchez-Villegas A, Toledo-Atucha EA, Faulin-Fajardo J. Bioestadística amigable. 3a ed. Barcelona: Elsevier; 2014. p.462-464).

  1. line 317, the denotation number was a typo? Was it “7” rather than “3”?

Yes, thank you for the observation, we have modified the error (line 368).

  1. line 369, the abbreviation “FI” should be defined clearly.

Thank you for your observation. We have deleted all the FI-abbreviations.

  1. line 401, the title of the Table 3 “…factor loadings for the of the…” should be deleted either “for the” or “of the.”

Thank you for your observation. We modified this and leaved “for the” (line 418).